# Vasohibin1, a new mouse cardiomyocyte IRES trans-acting factor that regulates translation in early hypoxia

**Fransky Hantelys[1†], Anne-Claire Godet[1†], Florian David[1†], Florence Tatin[1], Edith Renaud-Gabardos[1], Françoise Pujol[1], Leila H Diallo[1], Isabelle Ader[2], Laetitia Ligat[3], Anthony K Henras[4], Yasufumi Sato[5], Angelo Parini[1], Eric Lacazette[1], Barbara Garmy-Susini[1], Anne-Catherine Prats[1]\***

[1]UMR 1048-I2MC, Inserm, Université de Toulouse, UPS, Toulouse, France; [2]UMR 1031-STROMALAB, Inserm, CNRS ERL5311, Etablissement Français du Sang-Occitanie (EFS), National Veterinary School of Toulouse (ENVT), Université de Toulouse, UPS, Toulouse, France; [3]UMR 1037-CRCT, Inserm, CNRS, Université de Toulouse, UPS, Pôle Technologique-Plateau Protéomique, Toulouse, France; [4]UMR 5099-LBME, CBI, CNRS, Université de Toulouse, UPS, Toulouse, France; [5]Department of Vascular Biology, Institute of Development, Aging and Cancer, Tohoku University, Sendai, Japan

**Abstract** Hypoxia, a major inducer of angiogenesis, triggers major changes in gene expression at the transcriptional level. Furthermore, under hypoxia, global protein synthesis is blocked while internal ribosome entry sites (IRES) allow specific mRNAs to be translated. Here, we report the transcriptome and translatome signatures of (lymph)angiogenic genes in hypoxic HL-1 mouse cardiomyocytes: most genes are induced at the translatome level, including all IRES-containing mRNAs. Our data reveal activation of (lymph)angiogenic factor mRNA IRESs in early hypoxia. We identify vasohibin1 (VASH1) as an IRES trans-acting factor (ITAF) that is able to bind RNA and to activate the FGF1 IRES in hypoxia, but which tends to inhibit several IRESs in normoxia. VASH1 depletion has a wide impact on the translatome of (lymph)angiogenesis genes, suggesting that this protein can regulate translation positively or negatively in early hypoxia. Translational control thus appears as a pivotal process triggering new vessel formation in ischemic heart.

**\*For correspondence:**
anne-catherine.prats@inserm.fr

[†]These authors contributed equally to this work

**Competing interests:** The authors declare that no competing interests exist.

## Introduction

Hypoxia constitutes a major stress in different pathologies including both cancer and ischemic pathologies where artery occlusion leads to hypoxic conditions. In all of these pathologies, hypoxia induces a cell response that stimulates angiogenesis to re-feed starved cells with oxygen and nutrients (*Pouysségur et al., 2006*). Recently it has been shown that lymphangiogenesis is also induced by hypoxia (*Morfoisse et al., 2014*). Hypoxia-induced (lymph)angiogenesis is mediated by strong modification of gene expression at both transcriptional and post-transcriptional levels (*Pouysségur et al., 2006*; *Holcik and Sonenberg, 2005*). A major mode of regulation of gene expression is mediated at the transcriptional level by the hypoxia inducible factor 1 (HIF1), a transcription factor stabilized by oxygen deprivation, that activates transcription from promoters containing hypoxia-responsive elements (HRE). One of the well-described HIF1 targets is vascular endothelial growth factor A (VEGFA), a major angiogenic factor (*Forsythe et al., 1996*; *Pagès and Pouysségur, 2005*). However, two other major angiogenic or lymphangiogenic growth factors, fibroblast growth factor 2 (FGF2) and VEGFC, respectively, are induced by hypoxia in a HIF-independent

manner by a translational mechanism, indicating the importance of the post-transcriptional regulation of gene expression in this process (*Morfoisse et al., 2014*; *Conte et al., 2008*).

Translational control of gene expression plays a crucial role in the stress response. In particular, translation of mRNAs by the classical cap-dependent mechanism is silenced, whereas alternative translation mechanisms allow enhanced expression of a small group of messengers that are involved in the control of cell survival (*Holcik and Sonenberg, 2005*; *Baird et al., 2006*; *Spriggs et al., 2008*). One of the major alternative mechanisms that is able to overcome this global inhibition of translation by stress depends on internal ribosome entry sites (IRESs), which correspond to RNA structural elements that allow the direct recruitment of the ribosome onto mRNA. As regards the molecular mechanisms of IRES activation by stress, several studies have reported that RNA-binding proteins, called IRES trans-acting factors (ITAFs), are able to stabilize the adequate RNA conformation, thus allowing ribosome recruitment (*Faye and Holcik, 1849*; *Godet et al., 2019*; *Liberman et al., 2015*; *Mitchell et al., 2003*; *Morfoisse et al., 2016*). Interestingly, subcellular relocalization of ITAFs plays a critical role in IRES-dependent translation (*Lewis and Holcik, 2008*). Indeed, many RNA-binding proteins are known to shuttle between nucleus and cytoplasm, and it has been reported that cytoplasmic relocalization of ITAFs such as PTB, PCBP1, RBM4 or nucleolin is critical to activate IRES-dependent translation (*Godet et al., 2019*; *Morfoisse et al., 2016*; *Lewis and Holcik, 2008*). By contrast, other ITAFs such as hnRNPA1, may have a negative impact on IRES activity when accumulating in the cytoplasm (*Lewis et al., 2007*). However, how ITAFs participate in the regulation of the hypoxic response remains a challenging question.

IRESs are present in the mRNAs of several (lymph)angiogenic growth factors in the FGF and VEGF families, suggesting that the IRES-dependent mechanism might be a major way to activate angiogenesis and lymphangiogenesis during stress (*Morfoisse et al., 2014*; *Godet et al., 2019*; *Morfoisse et al., 2016*; *Huez et al., 1998*; *Martineau et al., 2004*; *Stein et al., 1998*; *Vagner et al., 1995*). However, most studies of the role of hypoxia in the regulation of gene expression have been performed in tumoral hypoxia, although it has been reported that tumoral angiogenesis leads to the formation of abnormal vessels that are non-functional, differing strongly from non-tumoral angiogenesis that induces formation of functional vessels (*Jain, 2005*). This suggests that the regulation of gene expression in response to hypoxia may be different in cancer versus ischemic pathologies. In particular, the role of IRESs in the control of gene expression in ischemic heart, the most frequent ischemic pathology, remains to be elucidated.

Here, we analyzed the transcriptome and the translatome of (lymph)angiogenic growth factors in hypoxic cardiomyocytes, and studied the regulation of IRES activities in early and late hypoxia. Data show that in cardiomyocyte, (lymph)angiogenic growth factors are mostly regulated at the translational level. Interestingly, IRESs of several mRNAs in the FGF and VEGF families are activated in early hypoxia in contrast to the IRESs of non-angiogenic messengers. We also looked for ITAFs governing IRES activation in hypoxia, and identified vasohibin1 (VASH1) as a new ITAF that is able to activate the FGF1 IRES in hypoxic cardiomyocytes but not the other IRESs studied here. VASH1 depletion has also a wide impact on the recruitment of (lymph)angiogenic factor mRNAs into polysomes, suggesting that this protein can regulate translation positively or negatively in early hypoxia.

## Results

### Most (lymph)angiogenic genes are not induced at the transcriptome level of hypoxic cardiomyocytes

In order to analyze the expression of angiogenic and lymphangiogenic growth factors in hypoxic cardiomyocytes, the HL-1 cell line was chosen: although immortalized, it keeps the beating phenotype specific to cardiomyocyte (*Claycomb et al., 1998*). HL-1 cells were submitted to increasing durations of hypoxia, from 5 min to 24 hr, and their transcriptome was analyzed on a Fluidigm Deltagene PCR array targeting 96 genes of angiogenesis, lymphangiogenesis and/or stress (*Figure 1*, *Figure 1—figure supplement 1*, *Supplementary file 1*). Data showed a significant increase of *Vegfa*, PAI-1 and apelin (*Apln*) mRNA levels, with a peak at 8 hr of hypoxia for *Vegfa* and PAI1 and 24 hr for *Apln*. These three genes are well-described HIF1 targets (*Forsythe et al., 1996*; *Kietzmann et al., 1999*; *Ronkainen et al., 2007*). However, although 56% of the detected genes were induced over 1.5-fold, few of them were strongly induced. Furthermore, the mRNA levels of several major angio- or

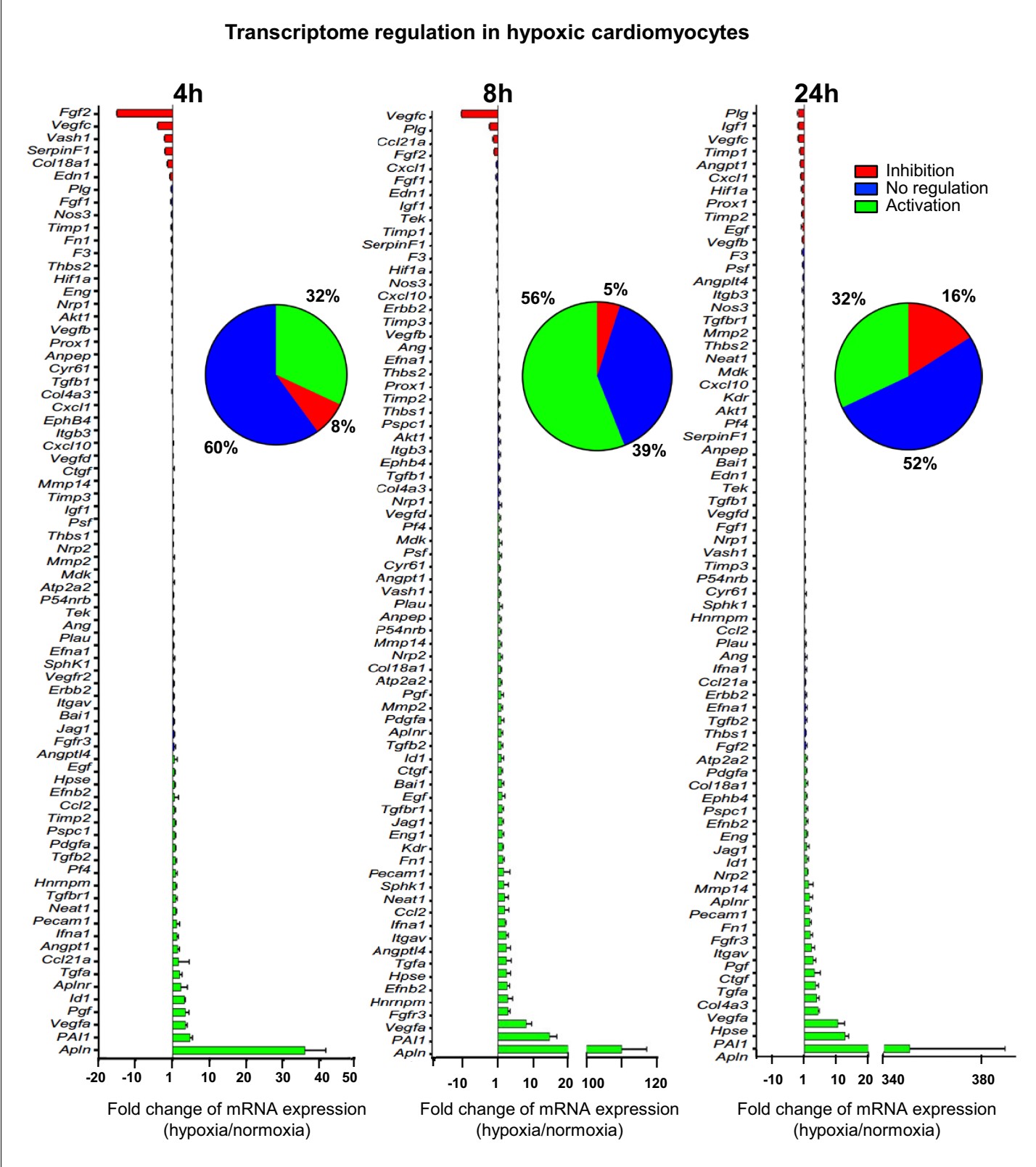

**Figure 1.** (Lymph)angiogenic genes are not drastically induced at the transcriptome level in hypoxic cardiomyocytes. Total RNA was purified from HL-1 cardiomyocytes submitted to increasing durations (from 5 min to 24 hr) of hypoxia at 1% $O_2$, as well as from normoxic cardiomyocytes as a control. cDNAs were synthesized and used for a Fluidigm deltagene PCR array dedicated to genes related to (lymph)angiogenesis or stress (*Supplementary file 6*). Analysis was performed in three biological replicates (cell culture well and cDNA), each of them measured in three technical

*Figure 1 continued on next page*

*Figure 1 continued*

replicates (PCR reactions). Relative quantification (RQ) of gene expression during hypoxia was calculated using the $2^{-\Delta\Delta CT}$ method with normalization to 18S rRNA and to normoxia. mRNA levels are presented as histograms for the times of 4 hr, 8 hr and 24 hr, as the fold change of repression (red) or induction (green) normalized to normoxia. Non-regulated mRNAs are represented in blue. The threshold for induction was set at 1.5. When the RQ value is inferior to 1, the fold change is expressed as $-1/RQ$. The percentage of repressed, induced, and non-regulated mRNAs is indicated for each duration of hypoxia. For shorter durations of 5 min to 2 hr, the percentages are shown in *Figure 1—figure supplement 1*. The detailed values for all of the durations of hypoxia are presented in *Supplementary file 1*.

The online version of this article includes the following figure supplement(s) for figure 1:

**Figure supplement 1.** Transcriptome of regulation in hypoxic cardiomyocytes.

lymphangiogenic factors, such as FGF2 and VEGFC, were strongly decreased after 4 hr or 8 hr of hypoxia. These data indicate that the transcriptional response to hypoxia in cardiomyocytes is not the major mechanism controlling the expression of (lymph)angiogenic factors, suggesting that post-transcriptional mechanisms are involved.

## mRNAs of most (lymph)angiogenic genes are recruited into polysomes in hypoxic cardiomyocytes

On the basis of the fact that mRNAs that are present in polysomes are actively translated, we tested the hypothesis of translational induction by analyzing the recruitment of mRNAs into polysomes. This experiment was performed in early and late hypoxia. The polysome profile showed that translational activity in normoxic HL-1 cells was low but decreased after 4 hr of hypoxia, with a shift of the polysome to monosome ratio from 1.55 to 1.40 (*Figure 2A*). Eukaryotic translation initiation factor 4E-binding protein 1 (4E-BP1) appeared as a single band and its phosphorylation profile did not change upon hypoxia, suggesting that this protein is already hypophosphorylated in normoxia in these cells (*Figure 2—figure supplement 1A and B*). By contrast, translation blockade was confirmed by the strong phosphorylation of eIF2α (*Figure 2B*, *Figure 2—figure supplement 1C*). 94% of the genes of the (lymph)angiogenic array showed a more sustained recruitment into polysomes under hypoxic conditions (*Figure 2C*, *Supplementary file 2*). This translational induction not only targets genes that encode major angiogenic factors and their receptors (e.g. *Vegfa*, *Fgf1*, *Pdgfa*, *Fgfr3*, *Vegfr2*), but also genes involved in cardiomyocyte survival in ischemic heart (*Igf1*, *Igf1r*) or in inflammation (BAI1, *Tgfb*). These data suggest that, in cardiomyocytes, the main response of (lymph) angiogenic genes to early hypoxia is not transcriptional, but translational.

## IRES-containing mRNAs are more efficiently mobilized into polysomes under hypoxic conditions

IRES-dependent translation has been reported to drive the translation of several mRNAs in stress conditions (*Morfoisse et al., 2014*; *Holcik and Sonenberg, 2005*; *Conte et al., 2008*; *Morfoisse et al., 2015*). Thus, we focused on the regulation of the different IRES-containing mRNAs present in the Fluidigm array (*Figure 3*). Interestingly, the only IRES-containing mRNA to be significantly induced by hypoxia at the transcriptome level was *Vegfa* (*Figure 3A* and *Figure 3—figure supplement 1*). Expression of the apelin receptor gene (*Aplnr*), which is presumably devoid of IRES but transcriptionally induced during hypoxia, is also shown for comparison.

The polysome recruitment of these IRES-containing mRNAs is shown in *Figure 3B*. Clearly, *Fgf1*, *Vegfa*, *Vegfd*, *Cyr61*, *Hif1a* and *Igf1r* mRNAs were recruited into polysomes under hypoxia, accumulating to levels 2 to 3 times those in normoxia, suggesting an important induction in terms of translation. By contrast, the recruitment of *Aplnr* mRNA into polysomes decreased about three times. The data are not available for *Fgf2* and *Vegfc* mRNAs, which were not detectable. These results indicate that hypoxia in cardiomyocytes, although blocking global cap-dependent translation, induces the translation of all detectable IRES-containing angiogenic factor mRNAs. This mechanism occurs as soon as 4 hr after oxygen deprivation, thus corresponding to an early event in the hypoxic response.

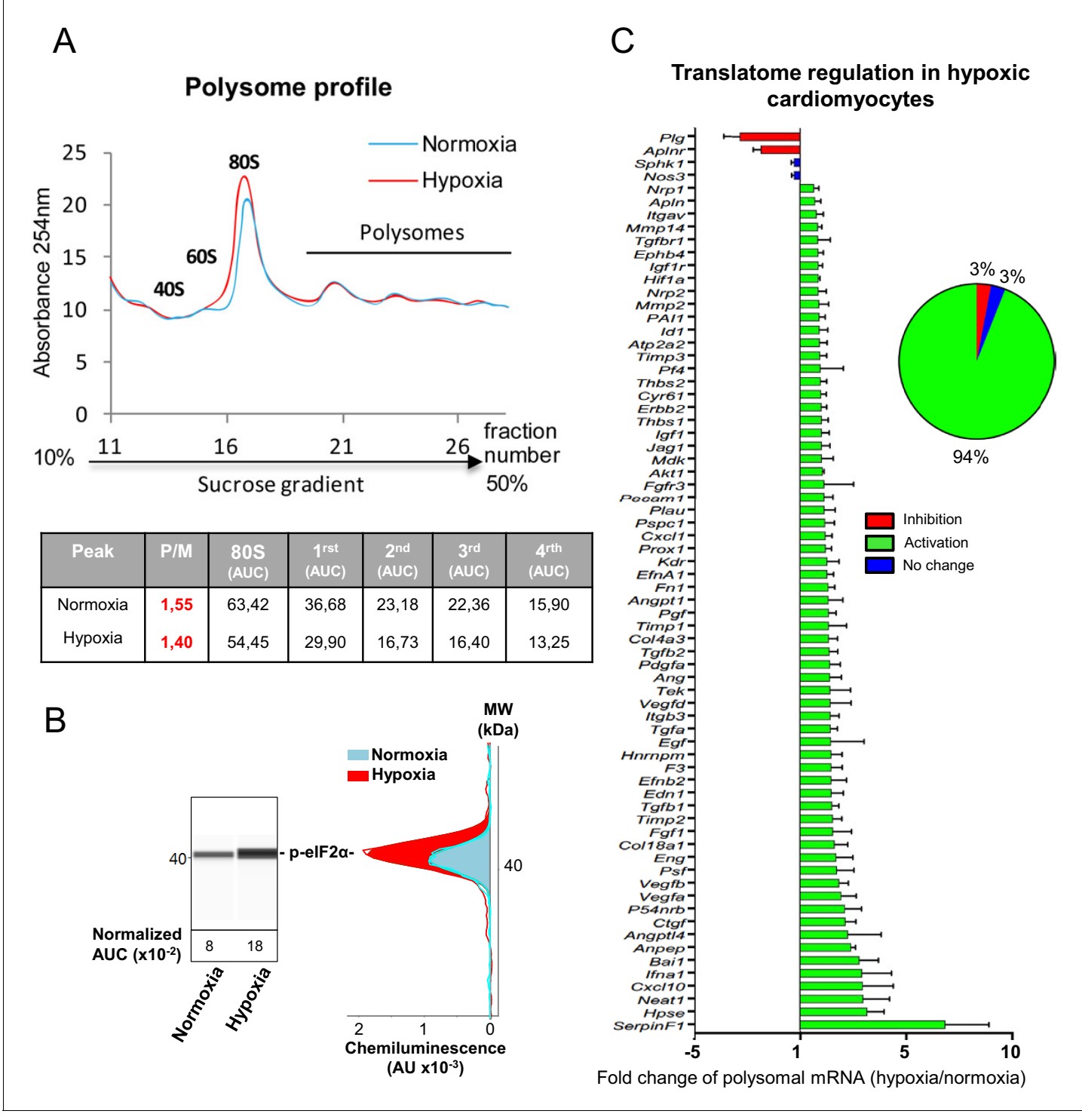

**Figure 2.** mRNAs of most (lymph)angiogenic genes are mobilized into polysomes in hypoxic cardiomyocytes. (**A-C**) In order to isolate translated mRNAs, polysomes from HL-1 cardiomyocytes in normoxia or after 4 hr of hypoxia at 1% $O_2$ were purified on a sucrose gradient, as described in 'Materials and methods'. Analysis was performed using a fluidigm PCR array from three biological replicates (cell culture well and cDNA), each of them measured in three technical replicates (PCR reactions). P/M ratio (polysome/monosome) was determined by delimiting the 80S and polysome peaks by taking the lowest plateau values between each peak and by calculating the area under the curve (AUC). Then the sum of area values of the four polysome peaks was divided by the area of the 80S peak (**A**). Translation blockade was measured by eIF2α phosphorylation quantification by Jess capillary Simple Western, normalized to the Jess quantification of total proteins (as described in 'Materials and methods'). Three independent experiments were done; a representative experiment is shown (**B**). RNA was purified from polysome fractions and from cell lysate before loading. cDNA

*Figure 2 continued on next page*

*Figure 2 continued*

and PCR arrays were performed as in *Figure 1*. Polysomes profiles are presented for normoxic and hypoxic cardiomyocytes. Relative quantification (RQ) of gene expression during hypoxia was calculated using the $2^{-\Delta\Delta CT}$ method with normalization to 18S rRNA and to normoxia. mRNA levels (polysomal RNA/total RNA) are shown as fold change of repression (red) or induction (green) in hypoxia normalized to normoxia as in *Figure 1C*. The threshold for induction was set at 1.5. When the RQ value is inferior to 1, the fold change is expressed as −1/RQ. The detailed values are available in *Supplementary file 2*.

The online version of this article includes the following figure supplement(s) for figure 2:

**Figure supplement 1.** Capillary electrophoresis immunodetection of 4E-BP1 and eIF2α.

## IRESs of (lymph)angiogenic factor mRNAs are activated during early hypoxia

To confirm that the polysome recruitment of IRES-containing mRNAs actually corresponds to a stimulation of IRES-dependent translation, IRESs from FGF and VEGF mRNAs were introduced into a bicistronic dual luciferase gene expression cassette (*Figure 4A*). Two IRESs from non-angiogenic mRNAs, c-myc and EMCV IRESs, were used as controls. A negative control without IRES was provided by a hairpin inserted between the two cistrons (*Créancier et al., 2000*). The well-established bicistronic vector strategy, previously validated by us and others, allows the measurement of IRES activity, which is revealed by expression of the second cistron, LucF (*Morfoisse et al., 2014*; *Créancier et al., 2000*). The bicistronic cassettes were subcloned into lentivectors because HL-1 cells are not efficiently transfected by plasmids but can be easily transduced by lentivectors, with an efficiency of more than 80% (not shown). HL-1 cardiomyocytes were first transduced with the lentivector containing the FGF1 IRES and the kinetics of IRES-dependent translation and protein expression were than examined after between 1 hr and 24 hr of hypoxia. Luciferase activities were measured from cell extracts and IRES activities were reported as the LucF/LucR luminescence ratio.

Data showed an increase in IRES activity between 4 hr and 8 hr of hypoxia whereas this activity decreased between 16 hr and 24 hr of hypoxia (*Figure 4B*). The expression of endogenous FGF1 was analyzed after 8 hr of hypoxia. FGF1 protein quantification (normalized to total proteins) showed that IRES induction correlates with an increased expression of FGF1 protein (*Figure 4C*). This is also consistent with the increase of FGF1 mRNA recruitment into polysomes observed above (*Figure 3B*, *Supplementary file 2*). To determine whether this transient induction could affect other IRESs, HL-1 cells were then transduced by the complete series of lentivectors described above (*Figure 4A*) and submitted to 4, 8 or 24 hr of hypoxia. Results showed an increase in all FGF and VEGF IRES activities in early hypoxia, except for VEGFA IRESa (4 hr and/or 8 hr), whereas the c-myc and EMCV IRESs were activated only in late hypoxia after 24 hr (*Figure 4D*). By contrast, VEGFA and VEGFD IRES activities decreased after 24 hr of hypoxia. The hairpin control was not induced (*Supplementary file 3J*). These data revealed two groups of IRESs: a first group including IRESs from (lymph)angiogenic growth factor mRNAs (except for VEGFA IRESa) that are activated during early hypoxia, and a second group including 'non-angiogenic' c-myc and EMCV IRESs, which are activated in late hypoxia.

## Identification of IRES-bound proteins in hypoxic cardiomyocytes reveals vasohibin1 as a new RNA-binding protein

Early activation of angiogenic factor IRESs during hypoxia suggested that specific ITAFs may be involved between 4 and 8 hr of hypoxia. In an attempt to identify such ITAFs, we used the biomolecular analysis coupled to mass spectrometry (BIA-MS) technology, which had been validated for ITAF identification in two previous studies (*Morfoisse et al., 2016*; *Ainaoui et al., 2015*). Biotinylated RNAs corresponding to FGF1 (early activation), VEGFAa (no activation) and EMCV IRESs (late activation) were used as probes for BIA-MS. Hooked proteins from normoxic and hypoxic HL-1 cells were then recovered and identified (*Figure 5A–B*, *Supplementary file 4*). Surprisingly, except for nucleolin bound to VEGFAa and EMCV IRES in normoxia, no known ITAF was identified as being bound to these IRESs in normoxia or in hypoxia. Interestingly, besides several proteins unrelated to (lymph)angiogenesis, we detected the presence of vasohibin1 (VASH1), a protein described as an endothelial cell-produced angiogenesis inhibitor and for its role in stress tolerance and cell survival (*Figure 5C*) (*Sato, 2012*; *Sato, 2015*). This secreted protein has never been reported to have any RNA-binding activity. VASH1 interaction with the FGF1 IRES was detected

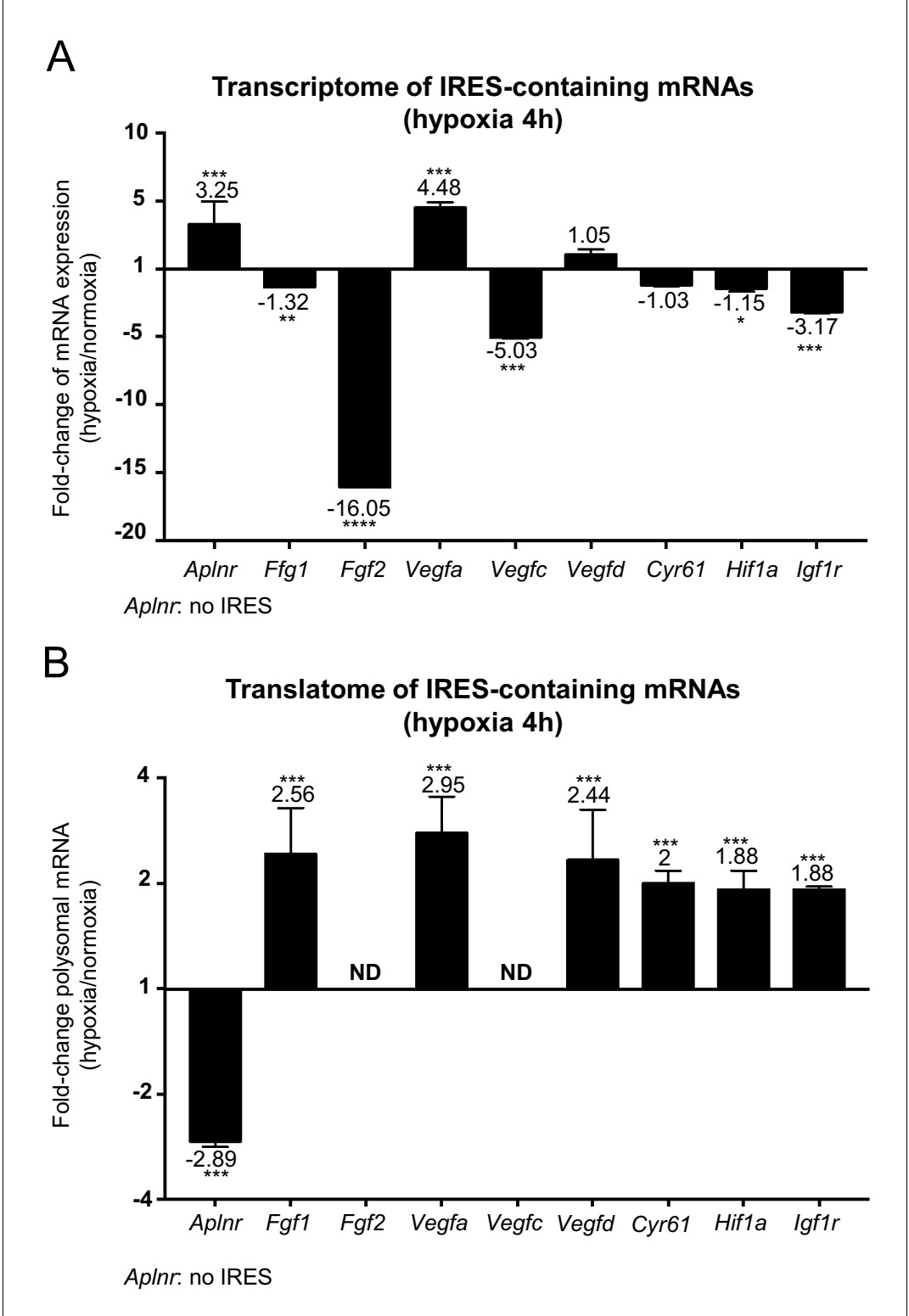

**Figure 3.** IRES-containing mRNAs are more efficiently associated with polysomes in hypoxic conditions. (**A, B**) RQ values for the IRES-containing mRNA transcriptome (**A**) and translatome (**B**) extracted from the PCR arrays shown in *Figures 1* and *2*. The gene *Aplnr* (apelin receptor) was chosen as a control without an IRES. *Vegfc* and *Fgf2* mRNAs, which are repressed in the transcriptome, were below the detection threshold in polysomes (ND). Histograms correspond to means ± standard deviation, with values for hypoxia compared to those for normoxia by a bilateral Student's test *p<0.05, **p<0.01, ***p<0.001, ****p<0.0001.
*Figure 3 continued on next page*

*Figure 3 continued*

The online version of this article includes the following figure supplement(s) for figure 3:

**Figure supplement 1.** Transcriptome of IRES-containing mRNAs in hypoxic cardiomyocytes.

after 4 hr or 8 hr of hypoxia, but not under normoxia (*Supplementary file 4*). This protein also interacted with the EMCV IRES both in normoxia and in hypoxia but not with the VEGFA IRES. In order to address the RNA-binding potential of VASH1, we performed an in silico analysis of VASH1 protein sequence that predicted two conserved RNA-binding domains (RBD) in the N- and C-terminal parts of the full-length protein, respectively (*Figure 5C*, *Figure 5—figure supplement 1A and B*). The direct interaction of VASH1 with FGF1, VEGFAa and EMCV IRESs was assessed by surface plasmon resonance using the full-length recombinant 44-kDa protein, resulting in the measurement of affinity constants of 6.5 nM, 8.0 nM and 9.6 nM, respectively (*Figure 5D–F*). These data indicate that VASH1 exhibits a significant RNA-binding activity.

## Vasohibin1 is translationally induced in early hypoxia and is localized in nuclear and cytoplasmic foci

VASH1 has been previously described for its expression in endothelial cells but never in cardiomyocytes (*Sato, 2012*). The present BIA-MS study provides evidence that it is expressed in HL-1 cardiomyocytes (*Supplementary file 4*). We analyzed the regulation of VASH1 expression during hypoxia: *Vash1* mRNA level strongly decreases after 4 hr of hypoxia whereas it is slightly upregulated after 8 hr (*Supplementary file 1*, *Figure 6A*). By contrast, analysis of *Vash1* mRNA recruitment into polysomes showed a strong increase at 4 hr of hypoxia (about 7-fold) (*Figure 6B*), whereas *Vash1* mRNA was not detectable in polysomes after 24 hr of hypoxia (*Supplementary file 2*). This indicates that *Vash1* mRNA translation is strongly induced in early hypoxia. This was confirmed by capillary Simple Western immunodetection, which showed that VASH1 protein expression increases after 4 hr of hypoxia (*Figure 6C*). Moreover, VASH1 subcellular localization was analyzed by immunocytochemistry: VASH1 appeared as foci in both cytoplasm and nucleus (*Figure 6D*). The number of foci did not change, but their size significantly increased in hypoxia (*Figure 6E and F*).

## Vasohibin1 is a new ITAF that is active in early hypoxia

The putative ITAF function of VASH1 was assessed by a knock-down approach using an siRNA smartpool (siVASH1). Transfection of HL-1 cardiomyocytes with siVASH1 was able to knock-down VASH1 mRNA with an efficiency of 73% (*Figure 7A*). The knock-down of VASH1 protein measured by capillary Western was only 59% (*Figure 7B*). This moderate knock-down efficiency was probably due to the long half-life of VASH1, superior to 24 hr (*Figure 7—figure supplement 1*). The effect of VASH1 knock-down was analyzed in HL-1 cells transduced with different IRES-containing bicistronic lentivectors in normoxia or after 8 hr of hypoxia. In normoxia, VASH1 knock-down generated a moderate increase of activity for several IRESs (13–16%), which was significant for VEGFD and EMCV IRESs (*Figure 7C*). By contrast, in hypoxia, VASH1 knock-down resulted in a strong decrease of FGF1 IRES activity, by 64%, whereas it did not significantly affect the other IRESs (*Figure 7D*). These data showed that VASH1 behaves as an activator of FGF IRES in hypoxia, whereas it tends to inhibit several IRESs in normoxia (*Figure 7C*).

## Vasohibin1 has a wide impact on the recruitment of (lymph) angiogenesis mRNAs into polysomes

In order to evaluate the possibility of a wider role for VASH1 in translational control, the (lymph) angiogenic factor mRNA polysome profile was analyzed in siVASH1-treated HL-1 cells in normoxia and after 8 hr of hypoxia (*Figure 8*, *Supplementary file 7*). VASH1 knock-down strongly affected the mobilization into polysomes of IRES-containing mRNAs, in normoxia and in hypoxia: *Vegfd* mRNA recruitment increased in normoxia, in concordance with the data shown in *Figure 7C*. *Igf1r* mRNA recruitment into polysomes decreased in hypoxia, whereas recruitment of *Hif1a* and *Vegfa* mRNAs increased. Unfortunately, *Ffg1* and *Vegfd* mRNAs were not detected in hypoxia, whereas *Fgf2* and *Vegfc* mRNAs were detected neither in normoxia nor in hypoxia in this array experiment, probably because they are poorly expressed. Globally, VASH1 depletion activated the polysome

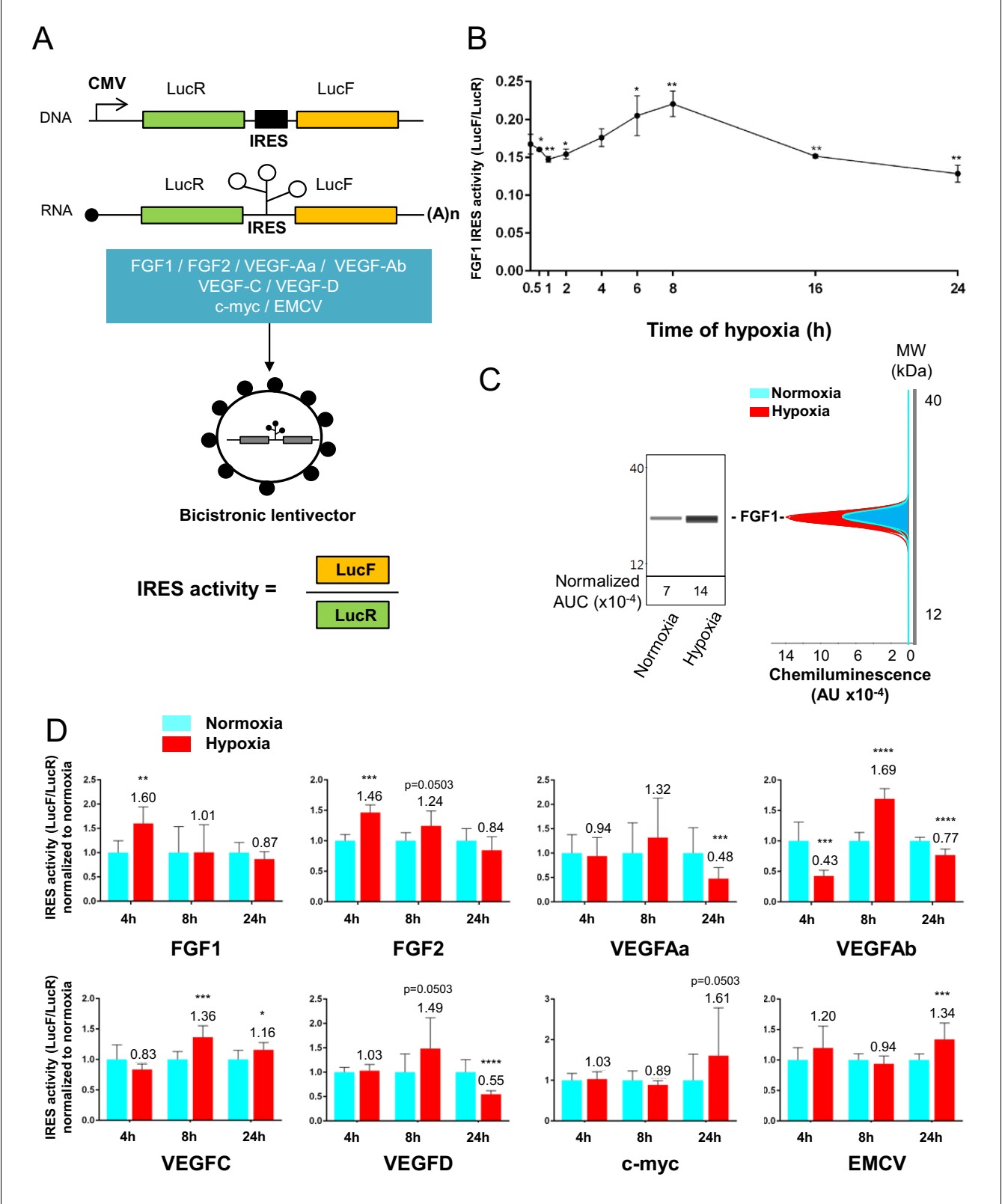

**Figure 4.** IRESs from (lymph)angiogenic factor mRNAs are activated in early hypoxia. (A–D) To measure IRES-dependent translation during hypoxia, HL-1 cardiomyocytes were transduced with bicistronic dual luciferase lentivectors (termed 'Lucky-Luke') containing different IRESs cloned between the genes of renilla (LucR) and firefly (LucF) luciferase (A). In bicistronic vectors, the translation of the first cistron LucR is cap-dependent, whereas translation of the second cistron LucF is IRES-dependent (*Créancier et al., 2000*). Cardiomyocytes transduced by the CRF1AL+ lentivector Lucky-Luke reporter

*Figure 4 continued on next page*

*Figure 4 continued*

containing FGF1 IRES were submitted to a hypoxia time-course (0 hr, 1 hr, 2 hr, 4 hr, 6 hr, 8 hr, 16 hr and 24 hr) and data from each time point were compared to those from normoxia with a non-parametric Mann-Whitney test (**B**). Endogenous FGF1 protein expression was measured by Jess capillary Simple Western of extracts of cardiomyocytes in normoxia or those submitted to 8 hr of hypoxia (normalized to Jess quantification of total proteins as described in 'Materials and methods'). Three independent experiments were performed; a representative experiment is shown (**C**). HL-1 cardiomyocytes transduced by different Lucky-Luke constructs were submitted to 4 hr, 8 hr or 24 hr of hypoxia and their luciferase activities were measured. IRES activities during hypoxia, expressed as LucF/LucR ratio, are normalized to those during normoxia. Histograms correspond to means ± standard deviation of the mean, data from hypoxic cardiomyocytes compared to those from normoxic cardiomyocytes with a non-parametric Mann-Whitney (M-W) test: *p<0.05, **p<0.01, ***<0.001, ****p<0.0001. For each IRES, the mean was calculated from nine cell culture biological replicates, each of them being the mean of three technical replicates (27 technical replicates in total but the M-W test was performed with n = 9). Detailed values of biological replicates are presented in *Supplementary file 3*. A no-IRES control was also used and values are presented in *Supplementary file 3J*.

recruitment of 22% and 44% of the detected mRNAs in normoxia and in hypoxia, respectively, whereas it inhibited the recruitment into polysomes of 41% versus 29% of detected mRNAs. Although this approach does not provide information about the mechanism of action, it strongly suggests a wide impact of VASH1, direct or indirect, on translation control. Furthermore, these data confirm that VASH1 has a dual role and can be either an activator or an inhibitor of translation.

## Discussion

The present study highlights the crucial role of translational control in cardiomyocyte response to hypoxia. Up to now, although a few genes had been described for their translational regulation by hypoxia, it was thought that most genes are transcriptionally regulated. Here, we show that translational control, revealed by mRNA recruitment into polysomes during hypoxia, regulates the majority of the genes involved in angiogenesis and lymphangiogenesis. IRES-dependent translation appears to be a key mechanism in this process, as we show that most of the (lymph)angiogenic mRNAs that are known to contain an IRES are upregulated during hypoxia. Furthermore, our data reveal that IRESs of angiogenic factor mRNAs are activated during early hypoxia, whereas the IRESs of non angiogenic mRNAs are activated in late hypoxia. We have identified an angiogenesis- and stress-related protein, VASH1, as a new ITAF that is responsible for the activation of the FGF1 IRES in early hypoxia, whereas it tends to inhibit other IRES activities in normoxia. VASH1 depletion has a positive or negative impact on the recruitment of many (lymph)angiogenesis mRNAs into polysomes, suggesting that this protein is widely involved, directly or not, in translational control in response to stress.

### Pathophysiological impact of a moderate stimulation of translation

A striking feature of our data is that the stimulation of IRES activities by hypoxia in cardiomyocytes is moderate, only by 1.3–1.7-fold. Are such small changes in growth factor expression sufficient to alter cellular programs? Several reports demonstrate that the answer is affirmative. An example is VEGFC IRES activation by hypoxia previously shown in tumor cells (*Morfoisse et al., 2014*). A 2–3-fold increase in endogenous VEGFC expression has a drastic effect on lymphatic vessel growth. In another study, a 2-fold increase in VEGFD IRES activity resulting from heat shock was sufficient to increase lymphatic vessel diameter (*Morfoisse et al., 2016*). The FGF1 IRES is activated by 1.7–2-fold during myoblast differentiation, and this is sufficient to promote myotube formation controlled by FGF1 (*Ainaoui et al., 2015*). We have also observed that cellular IRESs have often been reported previously to have a moderate activity in cell culture, whereas they can be much more active than a viral IRES and drastically regulated in vivo (*Créancier et al., 2000*). This can be explained by the use of cells that are immortalized and not in their physiological environment, which renders them less sensitive to stimuli than cells in vivo. Globally, cellular IRESs show a lower degree of activation than viral IRESs, as illustrated by *Braunstein et al. (2007)*, who report that the HIF1$\alpha$ IRES is stimulated by 1.6-fold during hypoxia, whereas the VEGFA IRES is stimulated by 2-fold and the EMCV IRESs by 3.5-fold.

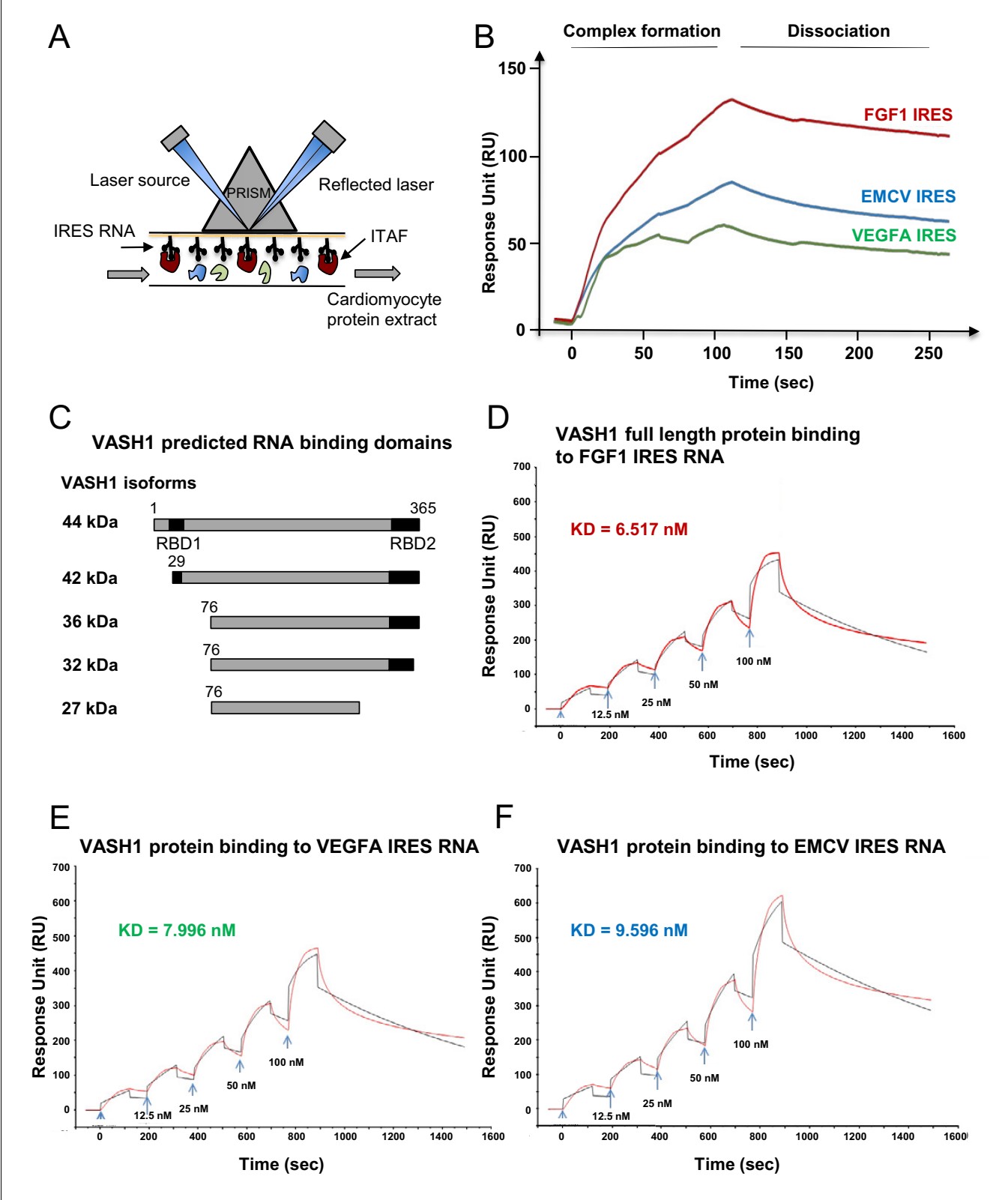

**Figure 5.** Identification of IRES-bound proteins in hypoxic cardiomyocytes reveals vasohibin1 as a new RNA-binding protein. (A–F) Biotinylated IRES RNAs were transcribed in vitro and immobilized on the sensorchip of the BIAcore T200 optical biosensor device (A). Total cell extracts from normoxic or hypoxic HL-1 cardiomyocytes were injected into the device. Complex formation and dissociation were measured (see 'Materials and methods') (B). Bound proteins were recovered as described in 'Materials and methods'. and identified by mass spectrometry (LC-MS/MS) after tryptic digestion. The *Figure 5 continued on next page*

*Figure 5 continued*

list of proteins bound in normoxia and hypoxia to FGF1, VEGFAa and EMCV IRESs is shown in *Supplementary file 4*. VASH1 protein was identified as being bound to FGF1 (hypoxia) and EMCV IRESs (hypoxia and normoxia), but not to VEGFA IRES. A diagram of VASH1 RNA-binding properties is shown, with VASH1 isoforms described by *Sonoda et al. (2006)* (C). The predicted RNA-binding domains (RBD1 and RBD2) shown in *Figure 5-figure supplement 1* that are conserved in mouse and human (C). Recombinant full-length 44-kDa VASH1 was injected into the Biacore T200 device containing immobilized FGF1 (D), VEGFAa (E) or EMCV (F) IRES as above. The affinity constants (KD) were calculated (D–F) with a single cycle kinetics (SCK) strategy.

The online version of this article includes the following figure supplement(s) for figure 5:

**Figure supplement 1.** Conservation of predicted RNA-binding domains in mouse and human vasohibin-1.

## Translational control in tumoral versus non-tumoral hypoxia

Most studies of gene expression in response to stress have been performed at the transcriptome level in tumoral cells of different origins, whereas the present study is focused on cardiomyocytes. HL-1 cardiomyocytes are immortalized but still exhibit the beating phenotype (*Claycomb et al., 1998*). Thus, this cell model, although not perfectly mimicking cardiomyocyte behavior in vivo, is still close to a physiological state. The strong translational response to hypoxia revealed by our data, which differs from the transcriptional response usually observed in tumor cells, may reflect mechanisms occurring in cells that are not engaged in the cell transformation process leading to cancer, or at least not too far along this process. Indeed, HL-1 cells respond to hypoxia very early, whereas the various murine or human tumor cell lines described in other reports require a longer period of hypoxia for IRES-dependent translation to be stimulated. In human breast cancer BT474 cells, VEGFA, HIF and EMCV IRESs are all activated after 24 hr of hypoxia (*Braunstein et al., 2007*). In murine 4T1 and LLC cells (breast and lung tumor, respectively), as well as in human CAPAN-1 pancreatic adenocarcinoma, the VEGFA and VEGFC IRESs are activated after 24 hr of hypoxia whereas the EMCV IRES is not activated (*Morfoisse et al., 2014*). The same observation of late activation in 4T1 cells has been made for the FGF1 IRES, whereas this IRES is activated in early hypoxia in HL-1 cardiomyocytes (AC Godet and AC Prats, unpublished data) (*Figure 4*). These observations suggest that many tumoral cell lines that develop resistance to hypoxia are not able to govern subtle the regulations of gene expression in early hypoxia observed in HL-1 cells.

## VASH1, an ITAF of early hypoxia

We also consider the hypothesis that the important process of translational regulation observed in our study may be cardiomyocyte-specific. In such a case, IRES-dependent translation would depend on cell-type-specific ITAFs as well as the early response to hypoxia. These results are of great importance in regard to the acute stress response in ischemic heart that is necessary for recovery. By contrast, a delayed chronic response is known to be deleterious for heart healing (*Silvestre et al., 2008*). In agreement with this hypothesis, VASH1 expression is cell-type-specific: described up to now as endothelial-specific, this protein is not expressed in tumoral cells (*Sato, 2012*). In the present study, we show that this cell-type specificity extends to cardiomyocytes. Consistent with our data, VASH1 has been described as a key actor in striated muscle angio-adaptation (*Kishlyansky et al., 2010*). This protein may thus have a role in the early hypoxic response in a limited number of cell types. The ITAF role of VASH1 identified here is physiologically relevant if one considers the function of VASH1 in angiogenesis and stress tolerance (*Sato, 2015*). According to previous reports, VASH1 is induced during angiogenesis in endothelial cells and halts this process, but its overexpression also renders the same cells resistant to senescence and cell death induced by stress (*Sato, 2015*). Furthermore, it has been reported that VASH1 is induced after 3 hr of cell stress at the protein level but not at the transcriptional level in endothelial cells (*Miyashita et al., 2012*). This is in agreement with our observation in cardiomyocytes where VASH1, although downregulated in the transcriptome in early hypoxia, is more efficiently recruited in polysomes at the same time (*Figure 6*).

It is noteworthy that VASH1 itself seems to be induced translationally by stress (*Figure 6*) (*Miyashita et al., 2012*). In endothelial cells, *Miyashita et al. (2012)* report that the protein HuR

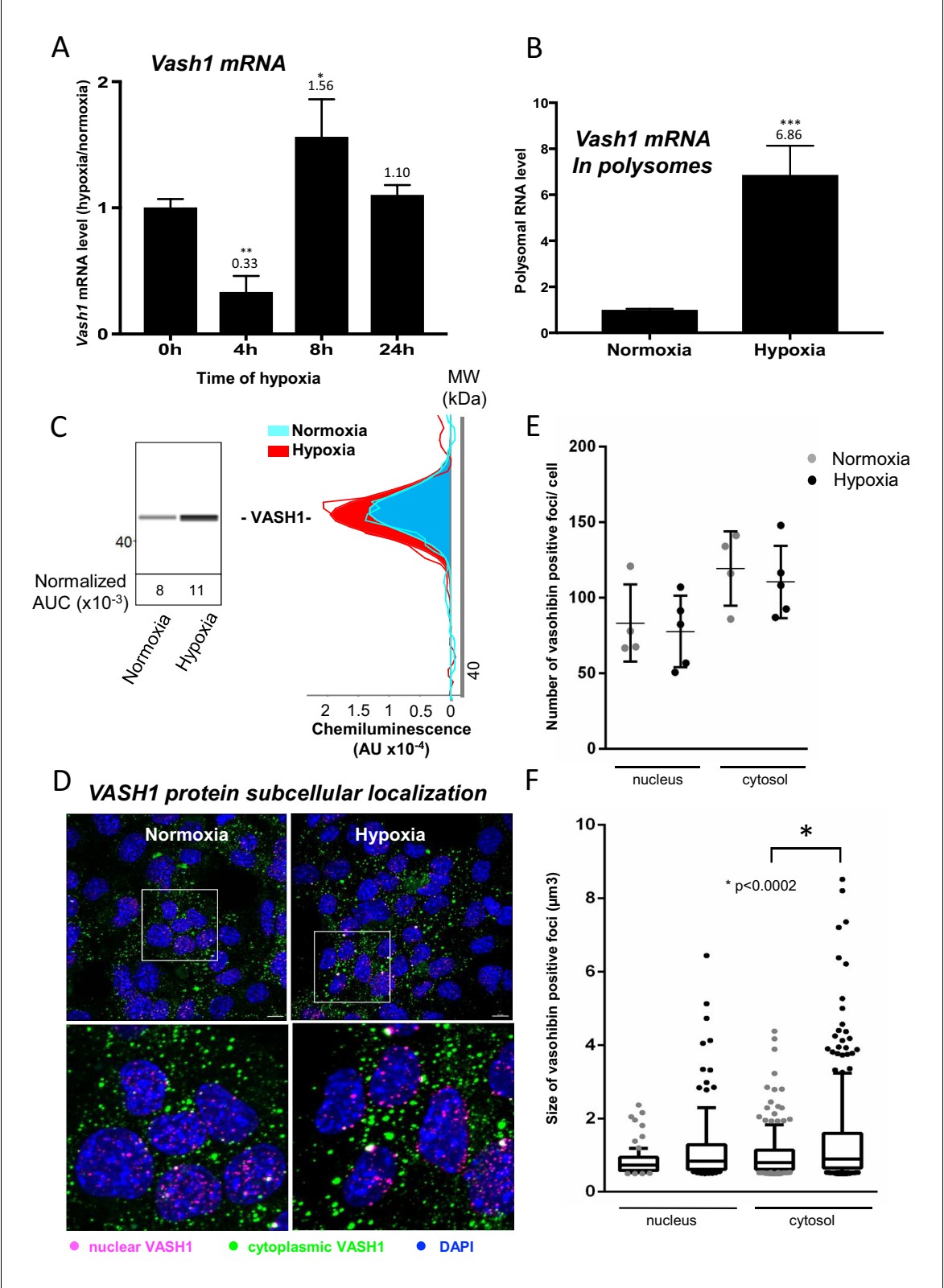

**Figure 6.** Vasohibin1 is translationally induced in early hypoxia and is localized in nuclear and cytoplasmic foci. (A–D) VASH1 expression was analyzed in HL-1 cardiomyocytes subjected to hypoxia at the transcriptome and translatome levels. A fluidigm RT qPCR array (*Supplementary file 2*) was performed with two biological replicates (cell culture and cDNA), each of them measured in two technical replicates (PCR reactions). Detailed values at 4 hr and 24 hr are presented in *Supplementary file 2*. As for *Figure 2*, total RNA was purified from the cell lysate of cardiomyocytes in normoxia or

*Figure 6 continued on next page*

*Figure 6 continued*

submitted to 4 hr, 8 hr or 24 hr of hypoxia (A). Polysomal RNA was purified from cardiomyocytes in normoxia or after 4 hr of hypoxia, from pooled heavy fractions containing polysomes (fractions 19–27) (B). Histograms correspond to mean ± standard deviation of the mean, with two-tailed t-test, *p<0.05, **p<0.01, ***<0.001 used to compare data from hypoxic and normoxic cardiomyocytes. VASH1 protein expression was measured by capillary Simple Western of extracts from cardiomyocytes in normoxia or submitted to 4 hr of hypoxia (C). VASH1 was immunodetected in HL-1 cardiomyocytes in normoxia or after 4 hr of hypoxia (D). DAPI staining allows to detect VASH1 nuclear localization (MERGE). VASH1 foci in the nucleus are shown in purple and those in the cytoplasm in green using Imaris software. The number of VASH1 foci was quantified in the nucleus and in the cytoplasm in normoxia and after 4 hr of hypoxia (n = 4–5 images with a total cell number of 149 in normoxia and 178 in hypoxia) (E). Boxplots of volume of vasohibin foci in normoxia and hypoxia (F). All foci above 0.5 μm$^3$ were counted. Whiskers mark the 10% and the 90% percentiles with the mean in the center. One-way Anova with Tukey's comparisons test was applied.

upregulates VASH1 by binding to its mRNA. HuR may bind to an AU-rich element present in the 3' untranslated region of the VASH1 mRNA. However, in other studies, HuR has also been described as an ITAF, thus it is possible that VASH1 itself may be induced by an IRES-dependent mechanism (*Godet et al., 2019*; *Durie et al., 2011*; *Galbán et al., 2008*).

The anti-angiogenic function of VASH1 may appear inconsistent with its ability to activate the IRES of an angiogenic factor. However, our data also suggest that VASH1 might be an activator or an inhibitor of (lymph)angiogenic factor mRNA translation. Such a double role may explain the unique dual ability of VASH1 to inhibit angiogenesis and to promote endothelial cell survival (*Sato, 2015*; *Miyashita et al., 2012*). This could result from the existence of different VASH1 isoforms of 44 kDa, 42 kDa, 36 kDa, 32 kDa and 27 kDa, resulting from alternative splicing and/protein processing (*Kishlyansky et al., 2010*; *Kern et al., 2008*; *Sato, 2013*; *Sonoda et al., 2006*). Interestingly, p42 and p27 are the main isoforms expressed in heart, where the p44 is undetectable (*Kishlyansky et al., 2010*; *Sonoda et al., 2006*). One can expect that the ITAF function is carried by p42, which contains the two predicted RNA-binding domains (*Figure 5C*, *Figure 5—figure supplement 1*). VASH1 has been observed in both the nucleus and the cytoplasm, and no striking nucleocytoplasmic relocalization is visible in response to hypoxia, whereas other ITAFs (such as hnRNPA1 or nucleolin) shuttle to the cytoplasm upon stress (*Godet et al., 2019*; *Morfoisse et al., 2016*; *Lewis et al., 2007*; *Cammas et al., 2007*; *Dobbyn et al., 2008*). Interestingly, VASH1, appears as foci whose size increases in hypoxia, suggesting that it could be partly translocated to stress granules. This translocation has been reported for other ITAFs, such as hnRNPA1 and polypyrimidine-tract-binding protein (PTB) (*Godet et al., 2019*; *Borghese and Michiels, 2011*; *Guil et al., 2006*).

## VASH1 impact on translational control can be positive or negative

Among the IRESs analyzed in the present study, the FGF1 IRES is the only one strongly regulated by VASH1 in hypoxia. However, VASH1 was also bound to the EMCV IRES in the BIA-MS experiment, and calculation of affinity constants does not reveal significant differences in affinity for FGF1, VEGFA or EMCV IRES. This apparent inconsistency finds an explanation if one considers the effect of VASH1 in normoxia: IRESs tend to be activated upon VASH1 depletion, significantly so for VEGFD and EMCV IRESs. Such data suggest that VASH1 binding is probably not specific to a given IRES, but instead that different VASH1 partners are recruited in the IRESome and result in positive or negative effects of this ITAF. The hypothesis of a dual role for VASH1 in translational control is confirmed by the effect of VASH1 depletion on the translatome: recruitment into polysomes is affected negatively or positively for 60–70% of mRNAs, both in normoxia and in hypoxia. Although the RNA-binding ability of VASH1 has been clearly shown in the present study, we cannot affirm that VASH1 impact is direct for all of these mRNAs. Nevertheless, a dual role of activator and inhibitor has been reported for more than ten other ITAFs. Our hypothesis thus remains that the key to the regulation of IRES activity by ITAFs is not RNA-binding specificity but rather IRESome multi-partner composition (*Godet et al., 2019*).

# Materials and methods

### Key resources table

| Reagent type (species) or resource | Designation | Source or reference | Identifiers | Additional information |
|---|---|---|---|---|
| Gene (firefly, *Photinus*) | Luc+ | Promega | | Modified firefly luciferase |
| Gene (*Renilla reniformis*) | LucR | Promega | | *Renilla* luciferase |
| Strain, strain background (*Escherichia coli*) | TOP 10 | Thermofisher Scientific | | Transformation-competent cells (genotype : F– mcr A Δ (mrr –hsd RMS–mcr BC) φ 80lac ZΔ M15 ΔlacX 74 rec A1 ara D139 Δ (araleu) 7697 gal U gal K rps L (StrR) end A1 nup G) |
| Cell line (*Homo sapiens*) | 293FT | Invitrogen | R700-07 | High-transfection performance for lentivector production |
| Cell line (*Homo sapiens*) | HT1080 | ATCC | CCL-121 | |
| Cell line (*Mus musculus*) | HL-1 | William C. Claycomb (*Claycomb et al., 1998*) | | Cardiomyocyte cell line with beating phenotype |
| Recombinant DNA reagent | pTRIP-CRHL+ | Dryad, *Supplementary file 8A* (*Morfoisse et al., 2014*) | | Bicistronic SIN lentivector construct with the CMV promoter controlling expression of LucR and Luc+ separated by an intergenic palindromic sequence |
| Recombinant DNA reagent | pTRIP-CRF1AL+ | Sequence available on Dryad, (*Martineau et al., 2004*; *Ainaoui et al., 2015*) | | Bicistronic SIN lentivector construct with the CMV promoter controlling expression of LucR and Luc+ separated by the human FGF1 IRES A |
| Recombinant DNA reagent | pTRIP-CRFL+ | Sequence available on Dryad, (*Créancier et al., 2000*) | | Bicistronic SIN lentivector construct with the CMV promoter controlling expression of LucR and Luc+ separated by the human FGF2 IRES |
| Recombinant DNA reagent | pTRIP-CRVAL+ | Sequence available on Dryad, (*Huez et al., 1998*) | | Bicistronic SIN lentivector construct with the CMV promoter controlling expression of LucR and Luc+ separated by the human VEGFA IRESa |
| Recombinant DNA reagent | pTRIP-CRVBL+ | Sequence available on Dryad, (*Huez et al., 1998*) | | Bicistronic SIN lentivector construct with the CMV promoter controlling expression of LucR and Luc+ separated by the human VEGFA IRESb |
| Recombinant DNA reagent | pTRIP-CRhVCL+ | Sequence available on Dryad, (*Morfoisse et al., 2014*) | | Bicistronic SIN lentivector construct with the CMV promoter controlling expression of LucR and Luc+ separated by the human VEGFC IRES |

*Continued on next page*

*Continued*

| Reagent type (species) or resource | Designation | Source or reference | Identifiers | Additional information |
|---|---|---|---|---|
| Recombinant DNA reagent | pTRIP-CRhVDL+ | Sequence available on Dryad, (**Morfoisse et al., 2016**) | | Bicistronic SIN lentivector construct with the CMV promoter controlling expression of LucR and Luc+ separated by the human VEGFD IRES |
| Recombinant DNA reagent | pTRIP-CRMP2L+ | Sequence available on Dryad, (**Nanbru et al., 1997**) | | Bicistronic SIN lentivector construct with the CMV promoter controlling expression of LucR and Luc+ separated by the human c-myc IRES |
| Recombinant DNA reagent | pTRIP-CREL+ | Sequence available on Dryad, (**Créancier et al., 2000**) | | Bicistronic SIN lentivector construct with the CMV promoter controlling expression of LucR and Luc+ separated by the EMCV IRES |
| Recombinant DNA reagent | *pCMV-dR8.91* | Addgene | | Trans-complementing plasmid containing the lentiviral protein genes (gag, pol, rev…) |
| Recombinant DNA reagent | pCMV-VSV-G | Addgene | #8454 | Trans-complementing plasmid containing VSV G envelope protein gene |
| Transfected construct (mouse) | Acell SMARTpool targeting VASH1 (siVASH1) | Dharmacon | | siRNA targeting VASH1 (**Supplementary file 8B**) |
| Transfected construct (mouse) | Control SMARTpool | Dharmacon | | Scramble siRNA (**Supplementary file 8B**) |
| Sequence-based reagent | Deltagene assay | Fluidigm Corporation, oligonucleotide sequences available in **Supplementary file 6** | | PCR array with 96 oligonucleotides primer couples |
| Antibody | Mouse monoclonal anti-VASH-1 | Abcam | EPR17420 | Jess Western 1:100 |
| Antibody | Mouse monoclonal anti-VASH-1 | Abcam | ab176114 | IC 1:50 |
| Antibody | Mouse monoclonal anti-FGF1-1 | Abcam | EPR19989 | Jess Western 1:25 |
| Antibody | Mouse monoclonal anti-p21 | Santa Cruz | sc-6546 (F5) | Jess Western 1:10 |
| Antibody | Rabbit polyclonal anti eIF2α | Cell Signaling Technology | 9721 | Jess Western 1:50 |
| Antibody | Mouse monoclonal anti-phospho-eIF2α | Cell Signaling Technology | 2103 | Jess Western 1:50 |
| Antibody | Rabbit polyclonal anti-4EBP-1 | Cell Signaling Technology | 9452 | Jess Western 1:50 |
| Antibody | Rabbit polyclonal anti-phospho-4EBP-1 | Cell Signaling Technology | 9451 | Jess Western 1:50 |
| Antibody | Secondary-HRP (ready to use rabbit 'detection module') | Protein Simple | DM-001 | Jess Western |
| Antibody | Donkey anti-rabbit alexa488 | Jackson Immunoresearch | 711-545-152 | Secondary antibody for IC |
| Software, algorithm | PRISM | Graphpad | | Statistical analysis |

## Lentivector construction and production

Bicistronic lentivectors coding for the *renilla* luciferase (LucR) and the stabilized firefly luciferase Luc+ (called LucF in the text) were constructed from the dual luciferase lentivectors described previously, which contained Luc2CP (*Morfoisse et al., 2014*; *Morfoisse et al., 2016*). The LucR gene used here is a modified version of LucR in which all the predicted splice donor sites have been mutated. The cDNA sequences of the human FGF1, -2, VEGFA, -C, -D, c-myc and EMCV IRESs were introduced between the first (LucR) and the second cistron (LucF) (*Vagner et al., 1995*; *Nanbru et al., 1997*; *Prats et al., 2013*). IRES sequence sizes are: 430 nt (FGF1), 480 nt (FGF2), 302 nt (VEGFAa), 485 nt (VEGFAb), 419 nt (VEGFC), 507 nt (VEGFD), 363 nt (c-myc), and 640 nt (EMCV) (*Morfoisse et al., 2014*; *Morfoisse et al., 2016*; *Huez et al., 1998*; *Martineau et al., 2004*; *Vagner et al., 1995*; *Nanbru et al., 1997*). The two IRESs of the VEGFA have been used and are called VEGFAa and VEGFAb, respectively (*Huez et al., 1998*). The hairpin negative control contains a 63 nt long palindromic sequence cloned between LucR and LucF genes (*Supplementary file 8A*). This control has been successfully validated in previous studies (*Morfoisse et al., 2014*; *Créancier et al., 2000*). The expression cassettes were inserted into the SIN lentivector pTRIP-DU3-CMV-MCS vector described previously (*Prats et al., 2013*). All cassettes are under the control of the cytomegalovirus (CMV) promoter. All vector sequences are available on Dryad (Ape format). Plasmid construction and amplification was performed in the bacteria strain TOP10 (Thermofisher Scientific, Illkirch Graffenstaden, France).

Lentivector particles were produced using the $CaCl_2$ method by tri-transfection with the plasmids pCMV-dR8.91 and pCMV-VSVG, $CaCl_2$ and Hepes-buffered saline (Sigma-Aldrich, Saint-Quentin-Fallavier, France) into HEK-293FT cells. Viral supernatants were harvested 48 hr after transfection, passed through 0.45 μm PVDF filters (Dominique Dutscher SAS, Brumath, France) and stored in aliquots at −80°C until use. Viral production titers were assessed on HT1080 cells with serial dilutions of a lentivector expressing green fluorescent protein (GFP) and scored for GFP expression by flow cytometry analysis on a BD FACSVerse (BD Biosciences, Le Pont de Claix, France).

## Cell culture, transfection and transduction

293FT (Invitrogen R700-07) and HT1080 (ATCC CCL-121) cells were provided by Invitrogen (Villebon sur Yvette, France) and ATCC (Manassas, VA, USA), respectively. The 293FT cell line is derived from human embryonic kidney cells transformed by the simian virus 40 (SV40) large T antigen. This cell line is ideal for the production of high titers of lentivectors. HT1080 is a human transformed line expressing activated N-ras oncogene. It was used only for lentivector titration.

The two cell lines were cultured in DMEM-GlutaMAX + Pyruvate (Life Technologies SAS, Saint-Aubin, France), supplemented with 10% fetal bovine serum (FBS), and MEM essential and non-essential amino acids (Sigma-Aldrich).

Mouse atrial HL-1 cardiomyocytes were a kind gift from Dr. William C. Claycomb (Department of Biochemistry and Molecular Biology, School of Medicine, New Orleans) (*Claycomb et al., 1998*). HL-1 cells are derived from a tumor of a transgenic mouse in which expression of the SV40 large T antigen was targeted to atrial cardiomyocytes. These highly differentiated cardiomyocyte HL-1 cells can be cultured and maintain their cardiac (beating) phenotype. The authentication method is observation of the beating phenotype. As soon as the phenotype is lost, it is necessary to start again with cells from an earlier passage.

All the cell lines were tested negative for mycoplasma contamination every three months. None of them is on the list of the commonly misidentified cell lines maintained by the International Cell Line Authentication Committee.

HL-1 cells were cultured in Claycomb medium containing 10% FBS, penicillin/streptomycin (100 μg/mL-100μg/mL), 0.1 mM norepinephrine, and 2 mM L-glutamine. Cell culture flasks were precoated with a solution of 0.5% fibronectin and 0.02% gelatin 1 hr at 37°C (Sigma-Aldrich). To keep the HL-1 phenotype, cell culture was maintained as previously described (*Claycomb et al., 1998*). For hypoxia, cells were incubated at 37°C at 1% $O_2$. HL-1 cardiomyocytes were transfected by siRNAs as follows: one day after being plated, cells were transfected with 10 nM of small interference RNAs from Dharmacon Acell SMARTpool, targeting VASH1 (siVASH1) or as a non-targeting siRNA control (siControl), using Lipofectamine RNAiMax (Invitrogen) according to the manufacturer's

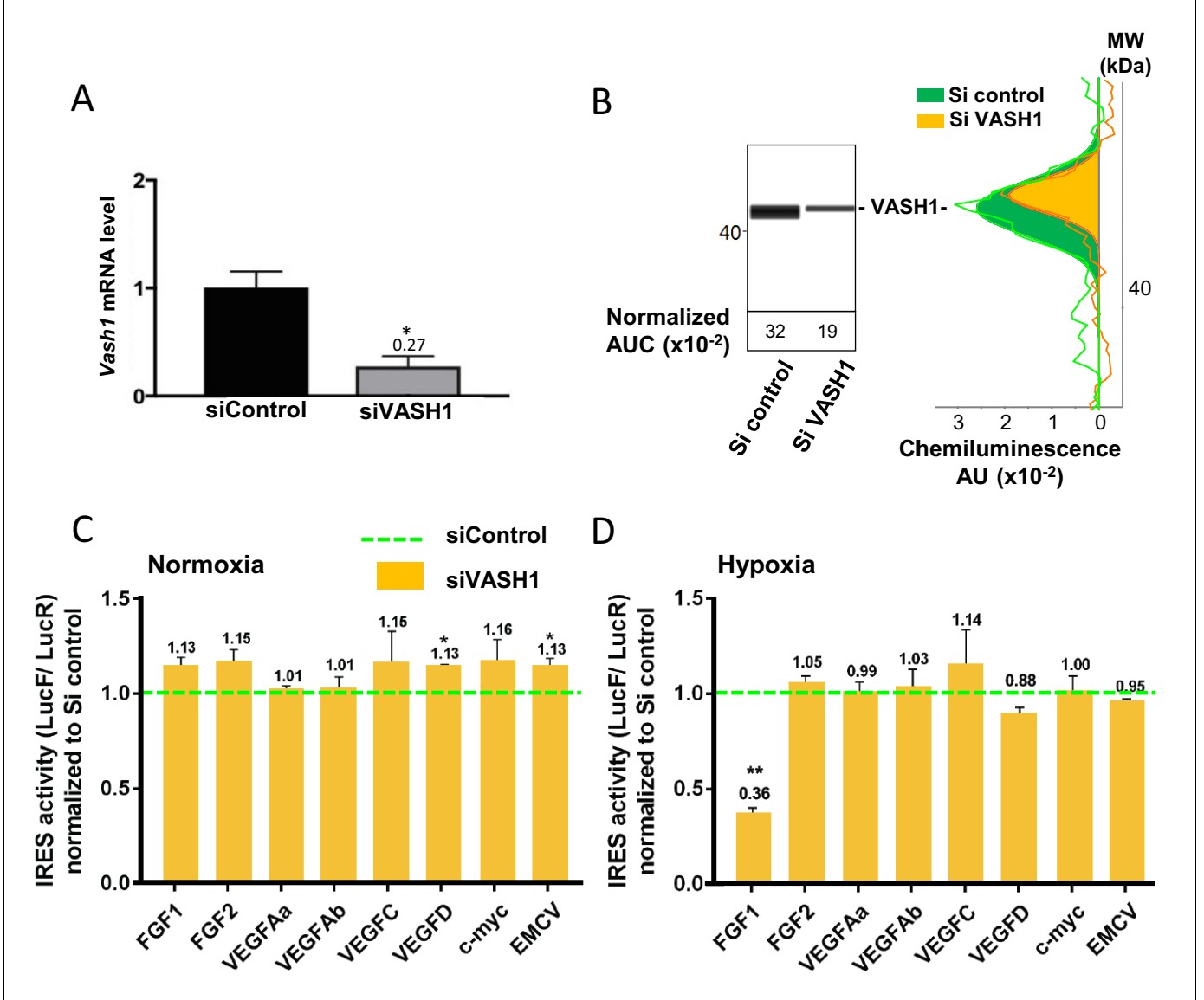

**Figure 7.** Vasohibin1 is a new ITAF that is active in early hypoxia. (A, B) VASH1 knock-down was performed in HL-1 cardiomyocytes using short-interfering (siRNA) smartpools targeting VASH1 (siVASH1) or control (siControl). VASH1 mRNA level was measured by RT-qPCR (A), and VASH1 protein expression was analyzed by the capillary Simple Western method using an anti-VASH1 antibody and quantified by normalization to total proteins. The experiments were reproduced twice, giving identical results. One of the two experiments is shown (B). Knock-down of VASH-1 was performed on cardiomyocytes transduced by a set of IRES-containing lentivectors used in *Figure 4*. (C, D) After 8 hr of hypoxia, IRES activities (LucF/LucR ratio) were measured in cell extracts from normoxic (C) and hypoxic cardiomyocytes (D). The IRES activity values have been normalized to the control siRNA. Histograms correspond to means ± standard deviation of the mean, and a non-parametric Mann-Whitney test was used to identify significant change from control levels: *p<0.05, **p<0.01. For each IRES the mean was calculated for nine cell culture biological replicates, each of these being the mean of three technical replicates (27 technical replicates in total but the M-W test was performed with n = 9). Detailed values of biological replicates are presented in *Supplementary file 5*.

The online version of this article includes the following figure supplement(s) for figure 7:

**Figure supplement 1.** VASH1 half-life is superior to 24 hr.

recommendations, in a media without penicillin-streptomycin and norepinephrine. Cells were incubated for 72 hr at 37°C with siRNA (siRNA sequences are provided in *Supplementary file 8B*).

For lentivector transduction, 6.10⁴ HL-1 cells were plated into each well of a six-well plate and transduced overnight in 1 mL of transduction medium (OptiMEM-GlutaMAX, Life Technologies SAS) containing 5 µg/mL protamine sulfate in the presence of lentivectors (MOI 2). A lentivector

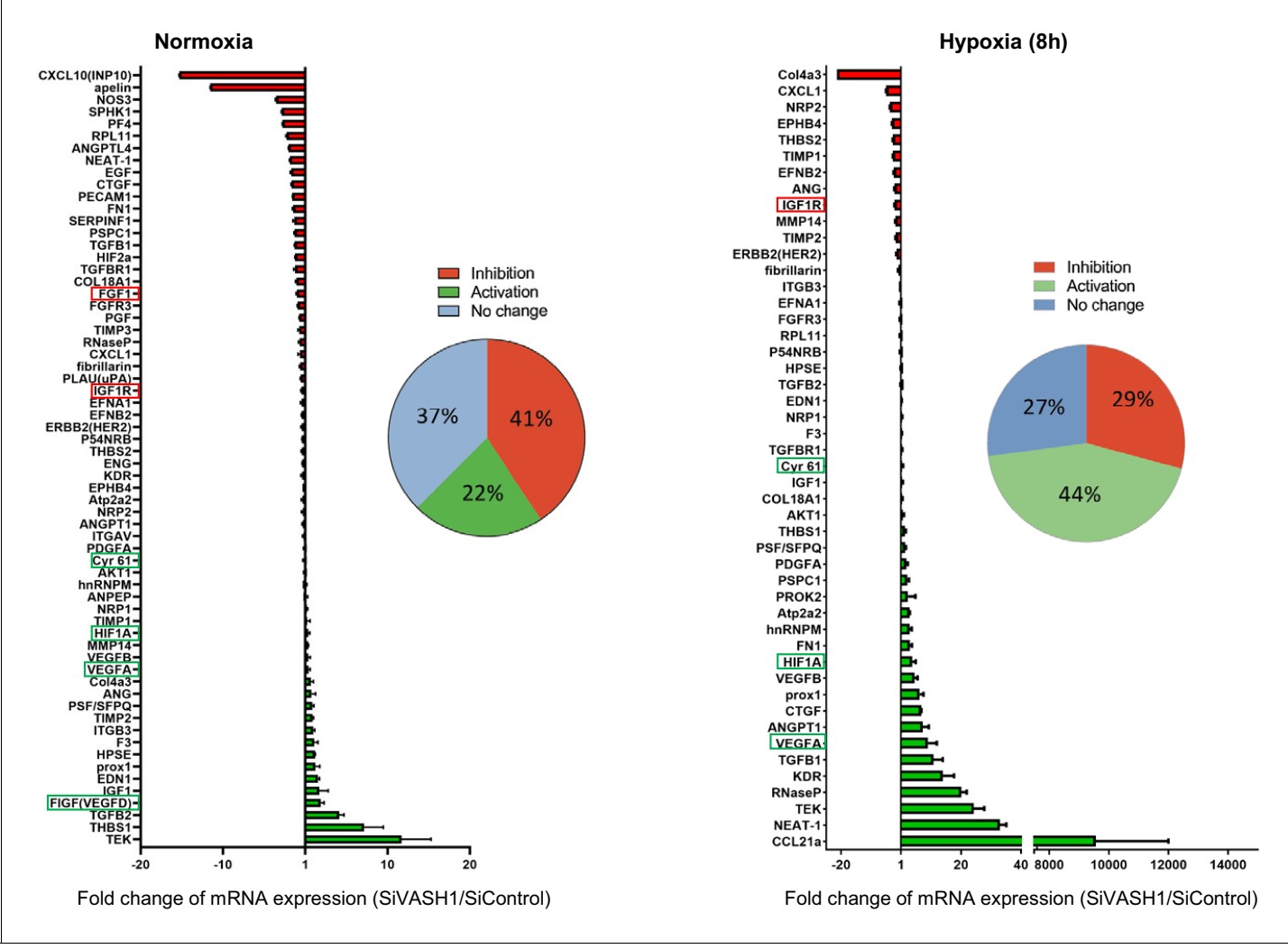

**Figure 8.** VASH1 depletion has both activating and inhibiting effects on mRNA recruitment into polysomes. HL-1 cardiomyocytes were treated with siVASH1 or siControl and submitted to 8 hr of hypoxia, or maintained in normoxia. RNA was purified from polysome fractions and from cell lysate before loading. cDNA and PCR arrays were performed as in *Figure 1*. Relative quantification (RQ) of gene expression during hypoxia was calculated using the $2^{-\Delta\Delta CT}$ method with normalization to 18S rRNA and to SiControl. mRNA levels (polysomal RNA/total RNA) are shown as fold change of repression (red) or induction (green) in siVASH1 cells normalized to SiControl-treated cells. Non-regulated mRNAs are represented in blue. The threshold for induction was set at 1.5. When the RQ value is inferior to 1, the fold change is expressed as $-1/RQ$. The detailed values are available in *Supplementary file 7*.

expressing GFP was used as a transduction control. GFP-positive cells were quantified 48 hr later by flow cytometry analysis on a BD FACSVerse (BD Biosciences). HL-1 cells were transduced with 80% efficiency. siRNA treatment of transduced cells was performed 72 hr after transduction (and after one cell passage).

To measure protein half-life, HL-1 cardiomyocytes were treated with cycloheximide (InSolution CalBioChem) diluted in PBS to a final concentration of 10 µg/mL in well plates. Time-course points were taken by stopping cell cultures after 0 hr, 4 hr, 6 hr, 8 hr, 16 hr or 24 hr of incubation and subsequent capillary Western analysis of cell extracts.

## Reporter activity assay

For reporter lentivectors, luciferase activities in vitro and in vivo were measured using a Dual-Luciferase Reporter Assay (Promega, Charbonnières-les-Bains, France). Briefly, proteins from HL-1 cells were extracted with Passive Lysis Buffer (Promega France). Quantification of bioluminescence was performed with a luminometer (Centro LB960, Berthold, Thoiry, France).

## Capillary electrophoresis and Jess simple western

Diluted protein lysate was mixed with fluorescent master mix and heated at 95°C for 5 min. 3 µL of protein mix containing protein normalization reagent, blocking reagent, wash buffer, target primary antibody (mouse anti-VASH-1 [Abcam EPR17420] diluted 1:100; mouse anti-FGF1 [Abcam EPR19989] diluted 1:25; mouse anti-P21 [Santa Cruz sc-6546 (F5)] diluted 1:10; rabbit anti eIF2α [Cell Signaling Technology 9721] diluted 1:50; mouse anti-phospho-eIF2α [Cell Signaling Technology 2103] diluted 1:50; rabbit anti-4EBP-1 [Cell Signaling Technology 9452] diluted 1:50; rabbit anti-phospho-4EBP-1 [Cell Signaling Technology 9451] diluted 1:50), secondary-HRP (ready-to-use rabbit 'detection module' [Protein Simple DM-001]), and chemiluminescent substrate were dispensed into designated wells in a manufacturer-provided microplate. The plate was loaded into the instrument (Jess, Protein Simple) and proteins were drawn into individual capillaries on a 25 capillary cassette (12–230 kDa) (SM-SW001). Data were analyzed using the compass software provided by the manufacturer. Normalization reagent allowed the detection of total proteins in the capillary through the binding of amine group by a biomolecule, and removed housekeeping proteins that can cause inconsistent and unreliable expression. No loading control is required with the Jess technology. The graphs in the figures show chemiluminescence values before normalization.

## RNA purification and cDNA synthesis

Total RNA extraction from HL-1 cells was performed using TRIzol reagent according to the manufacturer's instructions (Gibco BRL, Life Technologies, NY, USA). RNA quality and quantification were assessed using an Xpose spectrophotometer (Trinean, Gentbrugge, Belgium). RNA integrity was verified with an automated electrophoresis system (Fragment Analyzer, Advanced Analytical Technologies, Paris, France).

500 ng RNA was used to synthesize cDNA using a High-Capacity cDNA Reverse Transcription Kit (Applied Biosystems, Villebon-sur-Yvette, France). Appropriate no-reverse transcription and no-template controls were included in the PCR array plate to monitor potential reagent or genomic DNA contaminations, respectively. The resulting cDNA was diluted 10 times in nuclease-free water. All reactions for the PCR array were run in biological triplicates.

## qPCR array

The DELTAgene Assay was designed by the Fluidigm Corporation (San Francisco, USA). The qPCR-array was performed on BioMark with the Fluidigm 96.96 Dynamic Array, following the manufacturer's protocol (Real-Time PCR Analysis User Guide PN 68000088). The list of primers is provided in *Supplementary file 6*. A total of 1.25 ng of cDNA was preamplified using PreAmp Master Mix (Fluidigm, PN 100–5580, 100–5581; San Francisco, USA) in the plate thermal cycler at 95°C for 2 min, 10 cycles at 95°C for 15 s and 60°C for 4 min. The preamplified cDNA was treated by endonuclease I (New England BioLabs, PN M0293L; Massachusetts, USA) to remove unincorporated primers.

The preamplified cDNA was mixed with 2x SsoFast EvaGreen Supermix (BioRad, PN 172–5211; California, USA), 50 µM of mixed forward and reverse primers and sample Loading Reagent (Fluidigm, San Francisco, USA). The sample was loaded into the Dynamic Array 96.96 chip (Fluidigm San Francisco, USA). The qPCR reactions were performed in the BioMark RT-qPCR system. Data were analyzed using the BioMark RT-qPCR Analysis Software Version 2.0.

18S rRNA was used as a reference gene and all data were normalized on the basis of 18S rRNA level. *Hprt* was also assessed as a second reference gene but was not selected as its level was not stable during hypoxia. Relative quantification (RQ) of gene expression was calculated using the $2^{-\Delta\Delta CT}$ method. When the RQ value was inferior to 1, the fold change was expressed as $-1/RQ$. The oligonucleotide primers used are detailed in *Supplementary file 6*.

## Polysomal RNA preparation

HL-1 cells were cultured in 150 mm dishes. 15 min prior to harvesting, cells were treated with cycloheximide at 100 µg/ml. Cells were washed three times in PBS cold containing 100 µg/mL cycloheximide and scraped in the PBS/cycloheximide. After centrifugation at 3000 rpm for 2 min at 4°C, cells were lysed by 450 µl hypotonic lysis buffer (5 mM Tris-HCL [pH7.5]; 2.5 mM MgCl$_2$; 1.5 mM KCl). Cells were centrifuged at 13,000 rpm for 5 min at 4°C, before the supernatants were collected and loaded onto a 10–50% sucrose gradient. The gradients were centrifuged in a Beckman SW40Ti rotor

at 39,000 rpm for 2.5 hr at 4°C without brake. Fractions were collected using a Foxy JR ISCO collector and UV optical unit type 11. RNA was purified from pooled heavy fractions containing polysomes (fractions 19–27), as well as from cell lysate, before gradient loading.

## Preparation of biotinylated RNA

The FGF1, VEGFA or EMCV IRESs was cloned in pSCB-A-amp/kan plasmid (Agilent) downstream from the T7 sequence. The plasmids were linearized and in vitro transcription was performed with a MEGAscript T7 kit (Ambion), according to the manufacturer's protocol, in the presence of Biotin-16-UTP at 1 mM (Roche), as previously described (*Ainaoui et al., 2015*). The synthesized RNA was purified using an RNeasy kit (Qiagen).

## BIA-MS experiments

BIA-MS studies based on surface plasmonic resonance (SPR) technology were performed on a BIAcore T200 optical biosensor instrument (GE Healthcare), as described previously (*Morfoisse et al., 2016*; *Ainaoui et al., 2015*). Immobilization of biotinylated IRES RNAs was performed on a streptavidin-coated (SA) sensorchip in HBS-EP buffer (10 mM Hepes [pH 7.4], 150 mM NaCl, 3 mM EDTA, 0.005% surfactant P20) (GE Healthcare). All immobilization steps were performed at a flow rate of 2 μl/min with a final concentration of 100 μg/ml.

Binding analyses were performed with normoxic or hypoxic cell protein extracts at 100 μg/ml over the immobilized IRES RNA surface for 120 s at a flow rate of 30 μl/min. The channel (Fc1) was used as a reference surface for non-specific binding measurements. The recovery wizard was used to recover selected proteins from cell protein extracts. This step was carried out with 0.1% SDS. Five recovery procedures were performed to get amounts of proteins sufficient for MS identification.

Eluted protein samples from BIA experiment were digested in gel with 1 μg of trypsin (sequence grade, Promega) at 37°C. Peptides were then subjected to LC-MS/MS analysis. The peptide mixtures were loaded on a YMC-Triart C18 150 × 300 μm capillary column (particle diameter 3 μm) connected to a RS3000 Dionex HPLC system. The run length gradient (acetonitrile and water) was 30 min. Then, on the AB Sciex 5600+ mass spectrometer, data were acquired with a data-dependent analysis. Data were then loaded on Mascot software (Matrix Science) that attributes peptide interpretations to MS/MS recorded scans. The higher the score, the lower the probability of a false positive (a score of 20 corresponds to a 5% probability of a false positive).

## Surface plasmon resonance assays

For kinetic analysis, immobilization of biotinylated FGF1 IRES RNA was performed on a streptavidin-coated (SA) sensorchip in HBS-EP buffer (10 mM Hepes [pH 7.4], 150 mM NaCl, 3 mM EDTA, 0.005% surfactant P20) (GE Healthcare). The immobilization step was performed at a flow rate of 2 μl/min with a final concentration of 100 μg/ml. The total amount of immobilized FGF1 IRES RNA was 1500 RU.

Binding analyses were performed with recombinant protein VASH1 (Abnova H00022846-P01) at 100 μg/ml over the immobilized FGF1. This recombinant VASH1 contains the 27-kDa N-terminal part of the protein coupled to glutathione S-transferase. The channel (Fc1) was used as a reference surface for non-specific binding measurements.

A Single-Cycle Kinetics (SCK) analysis to determine association, dissociation and affinity constants (ka, kd, and KD respectively) was carried out by injecting different protein concentrations (16.25–300 nM). Binding parameters were obtained by fitting the overlaid sensorgrams with the 1:1. Langmuir binding model of the BIA evaluation software version 3.0.

## Immunocytology

Cells were plated on glass coverslips and incubated for 4 hr of normoxia or hypoxia. They were fixed with cold methanol at −20°C for 5 min, washed three times with PBS, and permeabilized for 1 min with 0.1% Triton. Then, cells were incubated for 5 min with blocking solution (1% FBS, 0.5% BSA) and 30 min with anti-VASH1 antibody (1/50; abcam ab176114) and Alexa 488 conjugated anti-mouse secondary antibody. Images were acquired with a LSM780 Zeiss confocal microscope, camera lens x60 with Z acquisition of 0.36 μM. A single plan is shown *Figure 6C*.

Imaris software was used to represent vasohibin staining in *Figure 6C*. To differentiate vasohibin in the nucleus and cytoplasm, the nucleus was delimitated with DAPI staining and all vasohibin foci in the nucleus are shown in purple, whereas those in the cytoplasm are shown in green.

Using Imaris software, the mean number of vasohibin foci was counted and the volume of vasohibin foci was quantified, a threshold was applied and all particles above $0.5\ \mu m^3$ were selected and quantified.

### Statistical analysis

All statistical analyses were performed using two-tailed Student's t-tests (*Figure 3*), Mann-Whitney tests (*Figure 4* and *Figure 7*) or one-way Anova with Tukey's comparisons test (*Figure 6*), *$p<0.05$, **$p<0.01$, ***$<0.001$, ****$<0.0001$. Data are expressed as mean ± standard deviation.

## Acknowledgements

Our thanks go to JJ Maoret and F Martins from the Inserm UMR1048 GeT-TQ plateau of the GeT platform Genotoul (Toulouse), F Lopez and L Tonini from the proteomic platform Genotoul (Toulouse), J Iacovoni from the Inserm UMR 1048 bioinformatics plateau, as well as L van den Berghe and C Segura from the Inserm UMR1037 vectorology plateau (Toulouse) and A Lucas from the We-Met Functional Biochemistry Facility (Toulouse). We also thank J Cavaille, C Müller and V Poinsot for helpful discussion and W Claycomb for providing HL-1 cells.

This work was supported by Région Midi-Pyrénées, Association Française contre les Myopathies (AFM-Téléthon), Association pour la Recherche sur le Cancer (ARC), European funding (REFBIO), Fondation Toulouse Cancer Santé and Agence Nationale de la Recherche ANR-18-CE11-0020-RIBO-CARD. FH received fellowships from the Région Midi-Pyrénées and from the Ligue Nationale Contre le Cancer (LNCC). ERG had a fellowship from AFM-Telethon. AC Godet had a fellowship from LNCC.

## Additional information

### Funding

| Funder | Grant reference number | Author |
| --- | --- | --- |
| Region Midi-Pyrenees | | Anne-Catherine Prats |
| AFM-Téléthon | | Edith Renaud-Gabardos<br>Anne-Catherine Prats |
| Association pour la Recherche sur le Cancer | | Anne-Catherine Prats |
| European Commission | REFBIO VEMT | Anne-Catherine Prats |
| Fondation Toulouse Cancer-Sante | | Barbara Garmy-Susini |
| Agence Nationale de la Recherche | ANR-18-CE11-0020-RIBOCARD | Anne-Catherine Prats |
| Ligue Contre le Cancer | | Fransky Hantelys<br>Anne-Claire Godet |

The funders had no role in study design, data collection and interpretation, or the decision to submit the work for publication.

### Author contributions

Fransky Hantelys, Conceptualization, Formal analysis, Investigation, Methodology; Anne-Claire Godet, Conceptualization, Data curation, Formal analysis, Investigation, Methodology; Florian David, Conceptualization, Investigation, Methodology; Florence Tatin, Formal analysis, Methodology; Edith Renaud-Gabardos, Resources, Formal analysis, Investigation; Françoise Pujol, Leila H Diallo, Investigation, Methodology; Isabelle Ader, Supervision, Investigation; Laetitia Ligat, Formal analysis,

Investigation, Methodology; Anthony K Henras, Supervision, Methodology; Yasufumi Sato, Resources, Formal analysis, Methodology; Angelo Parini, Conceptualization, Supervision, Funding acquisition; Eric Lacazette, Barbara Garmy-Susini, Conceptualization, Supervision, Methodology; Anne-Catherine Prats, Conceptualization, Data curation, Formal analysis, Supervision, Funding acquisition, Validation, Investigation, Visualization, Methodology, Project administration

### Author ORCIDs

Anne-Catherine Prats ⬢ https://orcid.org/0000-0002-5282-3776

### Decision letter and Author response

Decision letter https://doi.org/10.7554/eLife.50094.sa1
Author response https://doi.org/10.7554/eLife.50094.sa2

## Additional files

### Supplementary files

• Supplementary file 1. Transcriptome of (lymph)angiogenic factor genes in hypoxic HL-1 cardiomyocytes. Total RNA was purified from HL-1 cardiomyocytes submitted to increasing periods from 5 min to 24 hr of hypoxia at 1% $O_2$, as well as from normoxic cardiomyocytes as a control. cDNA was synthesized and used for a Fluidigm deltagene PCR array dedicated to genes related to (lymph)angiogenesis or stress (*Supplementary file 6*). Analysis was performed in three biological replicates (cell culture well and cDNA), each of them measured in three technical replicates (PCR reactions). Relative quantification (RQ) of gene expression in hypoxia was calculated using the $2^{-\Delta\Delta CT}$ method with normalization to 18S and to normoxia. Standard deviation is indicated. When the RQ value is inferior to 1, the fold change is expressed as $-1/RQ$. ND means 'non detected'. '–' means that the gene was not included in the array.

• Supplementary file 2. Translatome of (lymph)angiogenic factor genes in hypoxic HL-1 cardiomyocytes. Polysomes were purified on a sucrose gradient from HL-1 cardiomyocytes, either in normoxia or after 4 hr or 24 hr of hypoxia at 1% $O_2$, as described in 'Materials and Methods'. RNA was purified from polysome-bound fractions and from cell lysate (before gradient loading). cDNA and PCR arrays were performed as in *Figure 1* and in *Supplementary file 1*. Relative quantification (RQ) of gene expression in hypoxia was calculated using the $2^{-\Delta\Delta CT}$ method (polysomal RNA/total RNA normalized to normoxia). The 4 hr of hypoxia array was repeated in two independent arrays (RQ1 and RQ2). The values presented in *Figures 2* and *3* correspond to RQ1 values. In *Figure 6A and* B, values are from RQ2. For RQ1, gene expression analysis was performed in three biological replicates (cell culture well and cDNA), each of them measured in three technical replicates (PCR reactions). For RQ2 (4 hr and 24 hr), analysis was performed in two biological replicates, each of them measured in two technical replicates. Standard deviation is indicated. When the RQ value is inferior to 1, the fold change is expressed as $-1/RQ$. ND means 'non detected'. '–' means that the gene was not included in the array.

• Supplementary file 3. IRES activities after different periods of hypoxia in HL-1 cells. Luciferase activity values and IRES activities corresponding to the experiments presented in *Figure 4*. (A) Kinetics of FGF1 IRES activity from 30 min to 24 hr. (B–I) Activities of the different IRES after 4 hr, 8 hr and 24 hr of hypoxia. (J) Negative control with a lentivector containing a hairpin (no IRES) between the two luciferase cistrons. For each IRES and for each time, nine biological replicates were performed (n = 9). Each biological replicate corresponds to the mean of three technical replicates. Means, standard deviations (SD) and Mann-Whitney P values comparing IRES activities in hypoxia and in normoxia were calculated. The means are reported in the histograms shown in *Figure 4*. P-value significance is indicated: *p<0.05, **p<0.01, ***<0.001, ****p<0.0001.

• Supplementary file 4. BIA-MS analysis of IRES-bound proteins in hypoxic cardiomyocytes. (A–C) Total cell extracts from normoxic or hypoxic HL-1 cardiomyocytes were injected into the BIAcore T200 optical biosensor device where biotinylated IRES RNAs had been immobilized. The list of bound proteins identified by mass spectrometry (LC-MS/MS) after tryptic digestion is shown for FGF1 (A), VEGF-Aa (B) or EMCV (C) IRESs, respectively. The score and the number of spectra and

peptides identified are indicated. For each duration of hypoxia, cells were cultivated for the same period in normoxia as a control (normoxia 4 hr and 8 hr).

• Supplementary file 5. Knock-down of VASH1 in HL-1 cells. HL-1 cells transduced by the different IRES-containing lentivectors were transfected with siRNA SiVASH of SiControl and submitted to 8 hr of hypoxia. Luciferase activity and IRES activities (ratio LucF/LucR x 100) were measured. For each IRES, nine biological replicates were performed with SiVASH1 or SiControl (n = 9). Each biological replicate corresponds to the mean of three technical replicates. Means, standard deviations (SD) and Mann-Whitney P values comparing IRES activities with SiVASH1 or SiControl were calculated. IRES activities corresponding to the means of all biological replicates are reported in the histograms shown in *Figure 7*. P-value significance is indicated: *p<0.05, **p<0.01, ns = non-significant.

• Supplementary file 6. List of genes and primer couples used in the Fluidigm Deltagene PCR array.

• Supplementary file 7. VASH1 depletion has both activating and inhibiting effects on mRNA recruitment into polysomes. HL-1 cardiomyocytes were treated with siVASH1 of siControl and submitted to 8 hr of hypoxia or maintained in normoxia (see also *Figure 8*). RNA was purified from polysome fractions and from cell lysate before loading. cDNA and PCR array were performed as in *Figure 1*. Relative quantification (RQ) of gene expression during hypoxia was calculated using the $2^{-\Delta\Delta CT}$ method with normalization to 18S rRNA and to SiControl. mRNA levels (polysomal RNA/total RNA) are shown. When the RQ value is inferior to 1, the fold change is expressed as $-1/RQ$. 'ND' means that RNA was not detected.

• Supplementary file 8. Hairpin and siRNA sequences. (A) Sequence of the hairpin inserted in the bicistronic lentivector between the LucR and Luc+ genes. The LucR stop codon and the Luc+ start codon are indicated. The complementary sequences are indicated in red and in blue, respectively. (B) Sequences of the four siRNAs present in the siControl and siVASH1 smartpools.

• Transparent reporting form

## Data availability

All data generated or analysed during this study are included in the manuscript and supporting files. Lentivector plasmid complete maps and sequences are available on Dryad.

The following dataset was generated:

| Author(s) | Year | Dataset title | Dataset URL | Database and Identifier |
|---|---|---|---|---|
| Hantelys F, Godet A, David F, Tatin F, Renaud-Gabardos E, Pujol F, Diallo L, Ligat L, Henras A, Sato Y, Parini A, Lacazette E, Garmy-Susini B, Prats A, Ader I | 2019 | Data from: Vasohibin1, a new IRES trans-acting factor for induction of (lymph)angiogenic factors in early hypoxia | https://dx.doi.org/10.5061/dryad.2330r1b | Dryad Digital Repository, 10.5061/dryad.2330r1b |

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
