## [Decision Letter]

**Acceptance summary:**

The authors document the function of internal ribosome entry sites (IRESs) in mRNA translational control in cardiomyocytes under hypoxic conditions. They show that IRESes in several mRNAs encoding FGF and VEGF family members function in hypoxic cells. They also discovered that a surprisingly new IRES-trans-acting factor (ITAF) vasohibin (VASH1) is required for IRES-dependent translation, as VASH1 was reported earlier to exhibit anti-angiogenic activity in vivo in endothelial cells. Their paper suggests that the IRES-ITAF composition varies at different stages of hypoxia, engendering the sequential production of angiogenic factors required to form new functional vessels in the ischemic heart. The paper is highly relevant to the understanding of vascular biology, and pathologies in which hypoxia plays a major role, such as cardiovascular diseases and cancer.

**Decision letter after peer review:**

[Editors’ note: a previous version of this study was rejected after peer review, but the authors submitted for reconsideration. The first decision letter after peer review is shown below.]

Thank you for submitting your work entitled "Vasohibin1, a new IRES trans-acting factor for sequential induction of angiogenic factors in hypoxia" for consideration by *eLife*. Your article has been reviewed by a Senior Editor, a Reviewing Editor, and three reviewers. The reviewers have opted to remain anonymous.

Our decision has been reached after consultation between the reviewers. Based on these discussions and the individual reviews below, we regret to inform you that your work cannot be considered further for publication in *eLife*.

The three reviewers agree that the major problem is the lack of sufficient detail and rigor in this work and manuscript. The polysomes need to be quantified (P/M ratio) or repeated since the amount of lysate loaded is not identical. You should report raw numbers for normalized values, including statistical analysis and showing all data when possible rather than representative data. The manuscript is overly speculative given the data presented. The data do not provide sufficient support to the conclusion of waves of IRES activation or that Vash1 binding and relocalization are regulating IRES activity. These conclusions either have to be supported by more data or dramatically re-written. Thus, there is a significant amount of work that is required to render this paper accepted for publication in *eLife*. Because it is not clear whether the experiments can be performed in 2 months, we cannot accept the paper but would be willing to consider a new submission that successfully addresses all the issues raised.

Reviewer #1:

The manuscript by Prats et al., aims to better understand the role of IRESs in cardiomyocytes under hypoxic conditions. They show transcriptomic and polysome analysis that a lot of mRNAs that encode for proteins important for angiogenesis and lymph-angiogenesis are translationally regulated. They go on to show that many of these mRNAs have IRES activity, which is temporally regulated during hypoxia. Then they identify one ITAF, VASH1, that is required for FGF1 IRES activity. Overall, this is a very nice study that makes a significant contribution to our understanding of hypoxia in immortalized muscle cells. However, the biggest issue with this manuscript is the lack of details. Many of the experiments are not explained well enough in the methods or the figure legend for the reader to be confident that they are interpreting the results correctly.

Essential revisions:

It would be useful for the authors to justify why they chose 4, 8 and 24 hours for their time course. It would be nice to at least see how the IRES activity changed over a more detailed time course from 30 minutes to 24 hours having 10-15 time points. This would provide a better picture of the timing of the IRES regulation. For example, if hypoxia is translationally regulated why wouldn't we expect differences in 30 minutes or less? It would also be more compelling that there were differences in the timing of translation of these IRES containing mRNAs if there was more of a trend especially since the differences are so small (which doesn't mean that they aren't real). Also, given that the discussion focuses on how in many tumor cell lines that these IRESs are only activated after 24 hours of hypoxia, it would be informative to see if time points later than 24 hours resulted in a more robust activation of IRES activity. This would be necessary to support one of their major conclusions that the timing of IRES-mediated translation is regulated during hypoxia.

Figure 2A the authors claim that "The polysome profile displayed, as expected, a strong decrease of global translation (Figure 2A)." However, if there was a decrease in polysomes then there should be a corresponding increase in free subunits (40S and 60S), which is not seen. Rather their data show poor translational efficiency during normoxia and the reduced polysome peaks appear to be due to loading of less lysate. With different levels of lysate on the gradient the only definitive way to show that there is a significant reduction is to calculate the polysome (2-mer and up) to monosome (80S peak) ratio. The P/M should be shown for both normoxia and hypoxia polysomes.

In Figure 2 it is not clear how they calculated the fold change of the polysome associated mRNA under hypoxia. This is not adequately described in the figure legend and is not in the methods at all under Polysomal RNA preparation. Which fractions were collected for polysomes? Which fractions were free RNA? Whether all the fractions in the gradient were measured, which is important since equivalent levels of lysate was not used. How the statistics were calculated. The specific statistical test used should be in the figure legend rather than listing of all tests used in the manuscript in the methods. What is meant by "normalized to normoxia as in Figure 1" (which isn't explained in Figure 1 either).

In Figure 3 it is not clear again how the Fold-change in polysomal mRNA was calculated. Comparing it to polysome associated in normoxia? Or looking at the shift from monosomes to polysomes under hypoxic conditions. The first has issues with sample differences and these would need to be adequately addressed. The latter approach must be clearly detailed. Alternatively, if only the polysomes were examined then this would have issues with sample variation since generally polysomes are internally controlled and if only a fraction of the polysomes are assayed then it becomes impossible to know what fraction of the mRNA was polysome associated.

In Figure 4 they report that representative assays are shown. First the raw data needs to be reported in the supplementary material for all the replicates. This would include an empty vector with no IRES to show the background levels. Second, it is not clear why they are reporting representative data since it is a ratio of Fluc/Rluc which in theory should correct for differences in transduction efficiency. There is a significant concern that the differences observed are due to changes in Rluc values going down during hypoxia rather than a real increase in Fluc values. This should be cleared up by showing the raw data.

Table 2 the legend says "Relative quantification (RQ) of gene expression in hypoxia was calculated using the 2-ΔΔCT method (polysomal RNA/free RNA normalized to normoxia)." Does this mean that the data was normalized to normoxia either with or without siRNA KD of VASH1? Or was the KD data normalized to normoxia with VASH1 present in the cells? This should be clarified

The argument that the authors make for mRNAs that are translationally up-regulated by the VASH1 ITAF need VASH1 to re-localize to the nucleus to form the "IRESome". However, there is no data to show that preventing re-localization of VASH1 to the nucleus affects FGF1 IRES activity. It seems just as likely that the nuclear localization of the VASH1 may be for another function. Also, given the increase in VASH1 protein under hypoxia and its translational up-regulation, its overall nuclear to cytoplasmic distribution may remain unchanged but is more apparent in the nucleus at the higher expression levels. There is clearly VASH1 in the nucleus under normoxic conditions.

The authors should address why their reporter data does not match with their polysome data in that the reporter data shows the VEGFAa and VEGFAb IRESs were not increased above normoxic levels at 4 hours hypoxia (Figure 4C) whereas the polysome data shows they clearly are translationally upregulated (Figure 2 and Figure 3B). It is possible that they would see more dramatic changes in IRES activity of the reporters if they used a promoter such as the SV40 promoter, which is expressed at more physiological levels. There is some concern that using the higher expressing CMV promoter for these cellular IRESs reporters could saturate ITAFs therefore muting differences in IRES translation under hypoxic conditions for their reporters.

The methods for the real-time qPCR experiments is not clear. See the MIQE publication for best practices. We don't know how many replicates were measured, whether there were multiple dilutions of the sample measured, what the efficiency for the reactions were, which becomes critical given the small differences that were reported between samples. See "The MIQE Guidelines: Minimum Information for Publication of Quantitative Real-Time PCR Experiments".

Reviewer #2:

Hantelys and colleagues analyse changes in transcriptome and polysome association of mRNAs encoding lymphangiogenic factors during hypoxia. They observe differential activation of IRES-dependent translation along the hypoxic response and propose VASH1 as a protein binding and regulating the activity of certain IRESes. However, the manuscript at its present point is not convincing. There are several important controls missing in this study which are needed to support the conclusions:

1) Figure 2A: A decrease in polysomes should be accompanied by an increase in sub-polysomes or monosomal fractions. This does not seem to be the case of the profile shown here, suggesting that less material has been loaded in the gradient.

2) Table 3:

- What does the score mean and what would be a "good score"?

- Most proteins in this Table are detected with just one peptide. Proteins that are usual contaminants in affinity chromatography experiments (e.g. serum albumin) are also detected with one peptide here. How are true interactions distinguished from background?

- From the data in Table 3, the choice of VASH1 as a potential ITAF seems aleatory because of the low level of detection and the fact that there is no difference between 4 hours (where the FGF1 IRES is active) and 8 hours (where the IRES is inactive). This protein is detected with only one peptide in all instances, so the "selective binding depending on the IRES and stress condition" alluded to by the authors in the main text is difficult to believe.

3) Figure 5: The RNA binding constant of VASH1 for FGF1 IRES is somewhat high and the RBDs are only predicted. Both RNA binding and the assignment of VASH RBDs should be backed-up by controls. For example, how does binding to VEGF IRES looks like? Is binding to FGF1 IRES competed by FGF1 and EMCV IRESes but not by VEGF IRES? Does mutation of the predicted RBDs in VASH1 eliminate binding? Importantly, does VASH1 bind to endogenous FGF1 mRNA?

4) Figure 6: Please, show a Western blot with and without VASH1 depletion to ensure that VASH1 antibody only detects the expected protein(s). I do not see a perinuclear localization pattern, but rather a cytoplasmic pattern.

5) Figure 7C: Given that the activity of the various IRESes in the bicistronic assay is weak (2-fold maximum), it is important to ensure that the reduction after VASH1 depletion is significant and is not affected by the variability inherent to the assays. For this reason, rather than showing one representative triplicate experiment and use standard error of the mean to measure significance, it is important to use the full set of biological replicates and show standard deviation. The same could be applied to Figure 4B-D.

Also regarding Figure 7C, how do the authors explain the increase in VEGFA IRES activity upon VASH1 depletion if VASH1 does not bind to VEGFA IRES? The suggested inhibitory effect of VASH1 is more than discussible.

6) How does depletion of VASH1 affect the polysomal distribution of FGF1 and VEFGA mRNAs in hypoxia? Table 2 shows a comparison of polysomes in hypoxia (4 hours) with siVASH1 polysomes in normoxia, while the correct comparison would be siControl polysomes in hypoxia (4 hours) with siVASH1 polysomes in hypoxia (4 hours).

7) The Discussion section is highly speculative. The existence of "waves" of IRES-dependent regulation cannot be supported by the analysis of just 2-4 IRESes representing about 5% of the full set of translationally activated mRNAs under hypoxia. Furthermore, the physiological relevance of the potential regulation of IRES-dependent translation by VASH1 cannot be inferred by the established roles of VASH1, because each factor regulates multiple targets by different mechanisms. In addition, whether nuclear localization of VASH1 has any relevance for the function described here is an open question. The inference on regulons is too far-fetched given that there is only one member of each proposed regulon.

Reviewer #3:

In this manuscript the authors show that the FGF1 IRES functions in hypoxic cells, using a pathophysiological relevant system, and that vasohinbin1 is required for IRES-dependent translation. It is of interest that the authors have studied IRESs in models of hypoxia other than in tumour cells and the data provide new insights into how these regulatory RNA elements function. In general, the experiments have been well performed, but a couple of changes to the way in which the data are presented would help with the clarity of the MS, in addition to a small amount of additional experimentation.

Figure 2:

While the data show that there are certainly less "heavy" polysomes in the hypoxic cells it would be better if the areas under the curves were calculated so that these data can be presented additionally as a polysome:subpolysome ratio to provide a stronger indication of the degree of initiation changes. This would also allow the extent of variation between experimentation to be observed. In addition, the authors need to carry out Met labelling or use some other measure to show that global protein synthesis rates are decreasing under these conditions. This should be accompanied by western analysis to show changes in for example, 4EBPs, eIF2α.

Figure 3

The data show fold-changes in polysome mRNA localisation and in addition changes in transcription. Since the transcription and translation data have been obtained using arrays and, it is important to show how these transcripts vary across the gradients in hypoxic cells to confirm these findings, for a couple of mRNAs. Is it possible to calculate changes in translational efficiency from these data taking into account the transcriptional variation? This information would be very useful.

Figure 4

The data show how the IRES function changes, but it is important to show how these data correlate with changes in expression of the corresponding proteins. For example, is there an increase in VEGF of 1.9 fold after 8 hours of hypoxia? If the half-lives of these proteins are too long, pulse IP may need to be used.

Figure 6.

It is essential to measure the half-life of VASH1. Another explanation for the protein remaining expressed, even though the mRNA is decreasing, is that it is very stable with a long half-life. Therefore, it would not be turned over sufficiently during the course of the experiment to see a real difference.

[Editors’ note: what now follows is the decision letter after the authors submitted for further consideration.]

Thank you for submitting your article "Vasohibin1, a new IRES trans-acting factor for induction of (lymph)angiogenic factors in early hypoxia" for consideration by *eLife*. Your article has been reviewed by James Manley as the Senior Editor, a Reviewing Editor, and three reviewers. The reviewers have opted to remain anonymous.

The reviewers have discussed the reviews with one another and the Reviewing Editor has drafted this decision to help you prepare a revised submission.

Summary:

The authors investigated the role of IRESs in cardiomyocytes under hypoxic conditions. They show that the FGF1 IRES functions in hypoxic cells and that vasohinbin1 is required for IRES-dependent translation. They also show by polysome analysis that many mRNAs that encode for proteins important for angiogenesis and lymphangiogenesis are translationally regulated, many contain an IRES activity, which is temporally regulated during hypoxia.

Essential revisions:

The three original reviewers of your paper agree that you performed many additional important experiments and that the paper has improved considerably. However, as you can see two reviewers argue that some key issues remain to be addressed before the paper can be recommended for publication in *eLife*. These concerns were discussed and agreed on by all reviewers. In particular, it is critical to show that protein expression of endogenous VASH1 changes as the paper claims. You need to carry out pulse labeling IP of VASH1 to convincingly show changes in synthesis. You also ought to provide evidence that VASH1 specifically binds to the endogenous FGF-1 IRES. Another important issue that was raised by reviewer #2 concerns the statistical analysis of the data. The reviews of the three referees are attached so that you can read the detailed comments.

*Reviewer #1:*

This is an interesting study where it is shown that in hypoxic cells post-transcriptional control plays a major role in gene expression regulation. In particular the translation of IRES-containing mRNAs is upregulated and this is in part mediated by VASH1. The authors have carried out all additional experiments requested, and this has strengthened the manuscript. In my opinion this manuscript is now suitable for publication.

*Reviewer #2:*

This is a revised manuscript from Hantleys et al., entitled "Vasohibin1, a new IRES trans-acting factor for induction of (lymph)angiogenic factors in early hypoxia". The authors have been very responsive to the reviewer comments and the revised manuscript is significantly improved. Importantly, they showed the raw IRES luciferase values and included a control that would eliminate readthrough from the first to the second cistron (hairpin). Also, this negative control had 3 logs lower RLU values, which shows that they are measuring RLU activity well above background. The findings that are significant and novel include the study of IRESs in immortalized cells rather than transformed cells. Their work used immortalized cardiomyocytes to show that IRESs are induced during hypoxia. Interestingly, they found upregulation of hypoxic IRESs much earlier in these cell lines compared to what others have shown in transformed cells suggesting that cancer cells may be more resistant to hypoxia than normal cells. Yet, given that the cells they used are still immortalized and not primary cells, we still do not know how primary cells use IRESs during hypoxia. They show that several mRNAs become polysome associated following hypoxia and in particular they show that FGF1 IRES activity increases following induction of hypoxia. They also identify an ITAF, VASH1, that binds to IRESs in the nanomolar range and that FGF1 IRES activity depends on this ITAF during hypoxia.

However, there are a few things that reduce enthusiasm such as the Fgf1 western which doesn't look like there is an increase in protein levels under hypoxic conditions. Also, the IRES activities increase during hypoxia, but only by 1.2 to 1.7-fold. Is this sufficient enough for a biological response? Perhaps this is why the western didn't show a significant change in FGF1 protein levels or perhaps the increase in IRES activity is muted simply due to using immortalized cells or a promoter that overexpresses the dicistronic RNA. It is difficult to say. Even the ITAF, VASH1, which they reported bound to EMCV and FGF1 IRESs but not VEGFA IRES still had similar binding Kd for all of the IRESs. Overall, many of the conclusions are based on small effects or minor discrepancies that weaken the main conclusions, however, it is possible that these small changes in IRES activity are sufficient to alter cellular programs.

The authors reference a Figure 3C when there isn't a Figure 3C.

One of the key figures in the paper that supports a major conclusion that IRES activity leads to increased Fgf1 protein levels is not convincing. Figure 4C has no loading control, such as probing for another protein like actin that shouldn't change in amount from well to well (loading equal protein levels is standard but not sufficient). Furthermore, the quantification of the band intensities does not match the image of the bands in that there isn't an almost 2.5-fold increase in the Fgf1 band under hypoxic conditions (16 to 41). Nor does the graph in 4C (right) look like there is a 2-fold change. For example, figure EV 2 shows a 1.24-fold change that is striking so a 2.5-fold increase should very apparent. Overall, this figure was not clearly explained other than it is a western, thus it is not clear what the graph is or how to interpret this panel.

The hairpin control for the negative control for IRES activity is better than no control but it is not sufficient to rule out other artifacts such as cryptic promoter activity. One thing that would be very informative is to know whether the first and second cistron in the dicistronic reporter are in the same reading frame or not. Also, if the hairpin control has the ORFs in the same reading frame. The impression this reviewer gets is that the control is not really identical to the other IRES constructs, but a negative control used for other IRES constructs. Regardless, the hairpin IRES negative control is not explained in the methods. A true negative control would be the same reporter with an insert that didn't have IRES activity.

Nuclearized – used in Figure 6 legend and in the results should be changed to "relocalization to the nucleus" as nuclearized means to supply with nuclear weapons or deploy nuclear weapons in (such as outer space).

The description of "waves" of IRESs is not justified by the data as they did not clearly show that there are multiple IRESs that are activated at distinctly different times. It wasn't clear that there were 2 distinct classes of IRESs that were activated at 4 hours of hypoxia and a different class at 24 hours of hypoxia. Were there examples of IRES that were active at different times? Yes, but EMCV was up at 4 hours and 24 hours if their statistics are correct (Figure 4D). Also, the error bars were pretty large and the differences were not very big.

Reviewer #3:

The manuscript has improved with the new additions. The methods, raw data and corrections are more detailed, and this allows for a better understanding of the manuscript. In addition, the conclusions and Discussion section have been toned down. However, there are still numerous over-statements and several controls missing (some, by the way, said to be included).

Most importantly, the main conclusion of the paper, that VASH1 is an ITAF, is challenged by the lack of data showing that the protein indeed binds to endogenous IRES-containing mRNAs. The authors show that endogenous VASH1 binds to biotinylated IRESs, and that VASH1 affects the expression of one IRES reporter (that of FGF1), but no data are shown on the effect of VASH1 on endogenous transcripts or products. The effect on IRES reporters other than FGF1 is negligible, even if statistically significant for some. So VASH1 may very well be a functional ITAF for just one of the analyzed transcripts, without a generalized function in IRES-mediated translation or hypoxia.

Essential revisions:

1) For the reason stated above, I think the Title and Abstract are over-stated.

2) The authors conclude that most genes are not induced at the transcriptome level in hypoxic cardiomyocytes, while they are controlled at the polysome level. But the thresholds to consider induction are different in Figure 1 (transcriptomics) and Figure 2C (translatome).

3) Regarding Figure 1, an unbiased clustering to detect coordinated behavior of transcripts along hypoxia would be a good addition to the manuscript.

4) Figure 4C: The authors state that "IRES induction correlates with an increased expression of FGF1 protein". However, rather than an increase I see very similar bands and quantification peaks in this figure.

5) Figure 4D: The authors keep talking about activation "waves", when there is only one construct corresponding to a cellular mRNA (c-myc) that is activated at 24 hours, and where only a few IRESs (which the authors sustain represent all IRES-containing mRNAs involved in angiogenesis) have been analyzed altogether. This is another over-statement.

6) Figure 5: It is now clear that VASH1 binds RNA with high affinity, but the specificity is an issue. in vitro, the protein binds to all tested IRESs, and in the BIA-MS experiments VASH1 was detected 0/5 times bound to VEGFA, 2/5 times bound to FGF1, and 4/5 times bound to EMCV IRESs. The BIA-MS experiments were done in different conditions and were repeated just once per condition. From these data, the authors conclude that VASH1 shows specificity for FGF1 and EMCV in cellulo. In my view, additional assays are required to conclude that VASH1 shows specificity in cellulo, such as RIP-qPCR using oligos for endogenous mRNAs.

7) Figure 6: The authors state that "VASH1 immunodetection confirmed a strong expression of VASH1 at 4 hours of hypoxia, despite the decrease of its mRNA". They have not included these data in the manuscript. It is important that a Western blot showing the induction of the protein at 4hours of hypoxia compared to normoxia is included to back-up the polysomal RNA level analysis of part B.

Also in this figure, please use a term other than "nuclearized" to indicate an increase in size of VASH1 foci. This is a confusing term, as it implies the nuclear compartment.

8) Figure 7 shows that depletion of VASH1 affects FGF1 IRES activity under hypoxia. The effect on all other IRESs is negligible even if in some cases statistically significant. Therefore, I find the sentence "These data showed that VASH1 behaves as an activator ITAF in hypoxia, limited to FGF1, VEGFD and EMCV IRESs, while it has an inhibitory role on the activities of these IRESs in normoxia" a strong overstatement.

---

## [Author Response]

[Editors’ note: the author responses to the first round of peer review follow.]

Reviewer #1:The manuscript by Prats et al., aims to better understand the role of IRESs in cardiomyocytes under hypoxic conditions. They show transcriptomic and polysome analysis that a lot of mRNAs that encode for proteins important for angiogenesis and lymph-angiogenesis are translationally regulated. They go on to show that many of these mRNAs have IRES activity, which is temporally regulated during hypoxia. Then they identify one ITAF, VASH1, that is required for FGF1 IRES activity. Overall, this is a very nice study that makes a significant contribution to our understanding of hypoxia in immortalized muscle cells. However, the biggest issue with this manuscript is the lack of details. Many of the experiments are not explained well enough in the methods or the figure legend for the reader to be confident that they are interpreting the results correctly.

We fully agree about the lack of details and we have tried to answer this concern the best we can do. We have added explanations in the Material and methods section and in the figure legends. The details of our modifications are provided below.

Essential revisions:It would be useful for the authors to justify why they chose 4, 8 and 24 hours for their time course. It would be nice to at least see how the IRES activity changed over a more detailed time course from 30 minutes to 24 hours having 10-15 time points. This would provide a better picture of the timing of the IRES regulation. For example, if hypoxia is translationally regulated why wouldn't we expect differences in 30 minutes or less? It would also be more compelling that there were differences in the timing of translation of these IRES containing mRNAs if there was more of a trend especially since the differences are so small (which doesn't mean that they aren't real). Also, given that the discussion focuses on how in many tumor cell lines that these IRESs are only activated after 24 hours of hypoxia, it would be informative to see if time points later than 24 hours resulted in a more robust activation of IRES activity. This would be necessary to support one of their major conclusions that the timing of IRES-mediated translation is regulated during hypoxia.

We chose 4, 8 and 24 hours because usually hypoxia experiments are performed at 24 hours and we wanted to look at early hypoxia. In addition, expression of VEGFA mRNA (HIF-1 target) is highly increased after 8 hours (Figure 1 and EV Figure 3). At the translational level we have previously published that the FGF2 IRES is activated after 4 hours of hypoxia in another cell type (Conte et al., 2008). However, we agree that IRES-dependent translation could occur earlier. As suggested by the reviewer we performed the time course for the FGF1 IRES and finally found that the peak of activity (never tested before) is at 6 hours. This kinetics has been added in Figure 4A. Interestingly, the ratio LucF/LucR decreases at 1 hour, before starting to increase. We could not go after 24 hours as the cells do not resist.

Figure 2A the authors claim that "The polysome profile displayed, as expected, a strong decrease of global translation (Figure 2A)." However, if there was a decrease in polysomes then there should be a corresponding increase in free subunits (40S and 60S), which is not seen. Rather their data show poor translational efficiency during normoxia and the reduced polysome peaks appear to be due to loading of less lysate. With different levels of lysate on the gradient the only definitive way to show that there is a significant reduction is to calculate the polysome (2-mer and up) to monosome (80S peak) ratio. The P/M should be shown for both normoxia and hypoxia polysomes.

We agree that the polysome profile provided in the first version did not show any decrease of the P/M ratio and that less lysate had been loaded. We have reproduced the experiment, and a correct polysome profile is now shown, with a P/M ratio that decreases from 1.55 to 1.40. It was calculated as suggested by the reviewer, polysome (disome and up) to monosome. The translational efficiency is indeed poor in normoxia in these cells (often observed with non cancerous cells), but still decreases in hypoxia.

In Figure 2 it is not clear how they calculated the fold change of the polysome associated mRNA under hypoxia. This is not adequately described in the figure legend and is not in the methods at all under Polysomal RNA preparation. Which fractions were collected for polysomes? Which fractions were free RNA? Whether all the fractions in the gradient were measured, which is important since equivalent levels of lysate was not used. How the statistics were calculated. The specific statistical test used should be in the figure legend rather than listing of all tests used in the manuscript in the Materials and methods section. What is meant by "normalized to normoxia as in Figure 1" (which isn't explained in Figure 1 either).

Indeed we did not explain enough how we calculated the polysomal mRNA fold change: the RQ (Relative quantification) was measured by the 2^-DDCT^ method by calculating, for each mRNA, first the ratio of polysome-bound (disome fractions and up, fractions 19-27) to total RNA (from cell lysates before gradient loading), then the ratio hypoxia to normoxia which provided the fold change. When inferior to 1 (meaning a decrease), the RQ was expressed as -1/RQ to show the fold decrease. Then, the ratio polysome bound/total RNA was calculated to express the fold change in polysome recruitment. For more clarity, we have now explained this in Figure 2 legend and added all values in EV Table 2. The statistical test used in the manuscript is the Student test except for Figure 6 where we used ANOVA. We have added this information in each figure legend.

In Figure 3 it is not clear again how the Fold-change in polysomal mRNA was calculated. Comparing it to polysome associated in normoxia? Or looking at the shift from monosomes to polysomes under hypoxic conditions. The first has issues with sample differences and these would need to be adequately addressed. The latter approach must be clearly detailed. Alternatively, if only the polysomes were examined then this would have issues with sample variation since generally polysomes are internally controlled and if only a fraction of the polysomes are assayed then it becomes impossible to know what fraction of the mRNA was polysome associated.

Figure 3 just presents an extract from the data of Figure 1 and Figure 2 corresponding to the IRES-containing mRNAs. All the detailed values are in EV Table 1 for Figure 3A and in EV Table 2 for Figure 3A. The mode of calculation is also indicated in Figure 1 and Figure 2 legends and in EV Table 1 and Table 2 legends. For each mRNA, first the ratio of polysome-bound (disome fractions and up, pooled fractions 19-27) to total RNA (from cell lysates before gradient loading) was calculated, then the ratio hypoxia to normoxia, which provided the fold change. When inferior to 1 (meaning a decrease), the RQ was expressed as -1/RQ to show the fold decrease.

In Figure 4 they report that representative assays are shown. First the raw data needs to be reported in the supplementary material for all the replicates. This would include an empty vector with no IRES to show the background levels. Second, it is not clear why they are reporting representative data since it is a ratio of Fluc/Rluc which in theory should correct for differences in transduction efficiency. There is a significant concern that the differences observed are due to changes in Rluc values going down during hypoxia rather than a real increase in Fluc values. This should be cleared up by showing the raw data.

A table with the raw data (EV Table 3) has been provided for three independent experiments each including three biological replicates (n=9 for each IRES). An empty lentivector with an hairpin in place of an IRES (previously used in Morfoisse et al., 2014 by example) has been tested and provides LucF/LucR ratio lower than with all IRES and no significant difference betqeen normoxia and hypoxia)(EV Table 3J). The new Figure 4 shows the means of the 9 values). You can see in EV Table 3 that there is no decrease of LucR values at 4 and 8 hours, and that it only starts to decrease at 24 hours (and even not in all the experiments).

Table 2 the legend says "Relative quantification (RQ) of gene expression in hypoxia was calculated using the 2-ΔΔCT method (polysomal RNA/free RNA normalized to normoxia)." Does this mean that the data was normalized to normoxia either with or without siRNA KD of VASH1? Or was the KD data normalized to normoxia with VASH1 present in the cells? This should be clarified

EV table 2 has been modified for more clarity. The ΔCT values are presented for polysome bound and total RNA, as well as the RQ of bound/total ratio in normoxia and in hypoxia. Then the ratio of hypoxia/normoxia is presented, providing the fold change. We haved added the values for a second Fluidigm PCR array at 4 hours, and a PCR array at 24 hours. In each array, each gene was measured in triplicates. As regards the Fluidigm values with the siVASH1, we decided to remove them from the paper because they were not satisfactorily reproduced and thus, we could not really draw conclusions from them.

The argument that the authors make for mRNAs that are translationally up-regulated by the VASH1 ITAF need VASH1 to re-localize to the nucleus to form the "IRESome". However, there is no data to show that preventing re-localization of VASH1 to the nucleus affects FGF1 IRES activity. It seems just as likely that the nuclear localization of the VASH1 may be for another function. Also, given the increase in VASH1 protein under hypoxia and its translational up-regulation, its overall nuclear to cytoplasmic distribution may remain unchanged but is more apparent in the nucleus at the higher expression levels. There is clearly VASH1 in the nucleus under normoxic conditions.

Indeed, we admit that have no argument here showing that VASH1 has to be relocalized in the nucleus in hypoxia. In the revised version of Figure 6, we obtained higher quality pictures and performed quantification of the VASH1 positive foci in nucleus and in cytosol. Indeed, we clearly see VASH1 in the nucleus in normoxia and we do not measure any significant variation of the foci number between hypoxia and normoxia. However, the foci size clearly increases upon hypoxia. This suggests that newly synthesized VASH1 (shown by VASH1 mRNA recruitment in polysome Figure 6B) would localize in stress granules, as previously shown for other ITAFs such as hnRNPA1. Increase of stress granule size upon stress has been reported in a recent paper by Moon et al., 2019. As it is too early to show a model with the present data, we have removed the model proposed in the first version of Figure 7.

The authors should address why their reporter data does not match with their polysome data in that the reporter data shows the VEGFAa and VEGFAb IRESs were not increased above normoxic levels at 4 hours hypoxia (Figure 4C) whereas the polysome data shows they clearly are translationally upregulated (Figure 2 and Figure 3B). It is possible that they would see more dramatic changes in IRES activity of the reporters if they used a promoter such as the SV40 promoter, which is expressed at more physiological levels. There is some concern that using the higher expressing CMV promoter for these cellular IRESs reporters could saturate ITAFs therefore muting differences in IRES translation under hypoxic conditions for their reporters.

Indeed, we have no clear explanation for this difference. In the table with have never any activation at 4 hours for these two IRESs. The difference may come from a difference between the reporter and the endogenous mRNAs, or from a delay between the recruitment in polysomes and the luciferase accumulation. As we cannot explain this difference between 4 hours and 8 hours, we have modified our conclusion and we mention “early” hypoxia”, which includes 4 hours and 8 hours. We agree it would have been interesting to use with another promoter, as we have previously published ourselves that the promoter may affect IRES activity (Conte et al., 2009, Ainaoui et al., 2015), but it would have been a huge work to start again all these experiments (make all the constructs and lentivector production with another promoter, and perform all these experiments that are not easy with cardiomyocytes), whereas we observe significant data with the present promoter.

The methods for the real-time qPCR experiments is not clear. See the MIQE publication for best practices. We don't know how many replicates were measured, whether there were multiple dilutions of the sample measured, what the efficiency for the reactions were, which becomes critical given the small differences that were reported between samples. See "The MIQE Guidelines: Minimum Information for Publication of Quantitative Real-Time PCR Experiments".

We have added details in the Material and methods section about the replicates and dilutions, according to the MIQE guidelines.

Reviewer #2:Hantelys and colleagues analyse changes in transcriptome and polysome association of mRNAs encoding lymphangiogenic factors during hypoxia. They observe differential activation of IRES-dependent translation along the hypoxic response and propose VASH1 as a protein binding and regulating the activity of certain IRESes. However, the manuscript at its present point is not convincing. There are several important controls missing in this study which are needed to support the conclusions:1) Figure 2A: A decrease in polysomes should be accompanied by an increase in sub-polysomes or monosomal fractions. This does not seem to be the case of the profile shown here, suggesting that less material has been loaded in the gradient.

As mentioned above for reviewer 1, we agree that the polysome profile provided in the first version did not show any decrease of the P/M ratio and that less lysate had been loaded. We have reproduced the experiment, and a correct polysome profile is now shown, with a P/M ratio that decreases from 1.55 to 1.40. The translational efficiency is indeed poor in normoxia in these cells (often observed with non cancerous cells), but still decreases in hypoxia. Figure 2A has been modified and the new data have been included.

2) Table 3:- What does the score mean and what would be a "good score"?

The score reflects the probability for a peptide to be present. The higher the score, the lower the probability of false positive: by example a score of 20 corresponds to a 5% probability of false positive. Specifically, a peptide is considered as really present if the score is above 12-13%. A sentence has been added in the Materials and methods section.

- Most proteins in this Table are detected with just one peptide. Proteins that are usual contaminants in affinity chromatography experiments (e.g. serum albumin) are also detected with one peptide here. How are true interactions distinguished from background?

We would prefer having detected several peptides. However, the score of 28% allows to conclude that this peptide is present. In addition, it has been detected in several samples for FGF1 and EMCV IRESs.

- From the data in Table 3, the choice of VASH1 as a potential ITAF seems aleatory because of the low level of detection and the fact that there is no difference between 4 hours (where the FGF1 IRES is active) and 8 hours (where the IRES is inactive). This protein is detected with only one peptide in all instances, so the "selective binding depending on the IRES and stress condition" alluded to by the authors in the main text is difficult to believe.

The peak of IRES activity is between 4 hours and 8 hours but can move within this window depending on the cell batch and the differentiation state of the cells. HL-1 cells are very sensitive to early hypoxia when they are beating but it is difficult to maintain them in this beating state. In the kinetics performed more recently with a new cell batch, the peak is around 6 hours of hypoxia. In the revised version we consider only early hypoxia (4 hours to 8 hours) versus late hypoxia (24 hours).

Another explanation is that the BIA-MS approach does not take into account all the cell parameters, in particular VASH1 intracellular localization. In addition, the complex with RNA is formed in vitro and not in cellulo. Indeed, there is no binding specificity in vitro but it may result of the absence of other IRESome partners. We have removed sentences about selective binding. However, we can clearly conclude about the RNA binding feature of VASH1 (Figure 5D-F).

Only one VASH1 peptide was identified, but in several samples (FGF1 4 hours and 8 hours of hypoxia, EMCV normoxia and hypoxia) and with a significant score. As mentioned above, the higher the score, the lower the probability of false positive: by example a score of 20 corresponds to a 5% probability of false positive.

VASH1 has been logically selected among the different bound proteins because of its involvement in angiogenesis and in stress tolerance (Sato and Atheroscler Thromb, 2015). It seemed apparent to us that VASH1, as a physiolocally relevant candidate, was the best choice to start the ITAF study. However, the other candidates remain interesting for the continuation of the project.

3) Figure 5: The RNA binding constant of VASH1 for FGF1 IRES is somewhat high and the RBDs are only predicted. Both RNA binding and the assignment of VASH RBDs should be backed-up by controls. For example, how does binding to VEGF IRES looks like? Is binding to FGF1 IRES competed by FGF1 and EMCV IRESes but not by VEGF IRES? Does mutation of the predicted RBDs in VASH1 eliminate binding? Importantly, does VASH1 bind to endogenous FGF1 mRNA?

These first binding experiments were done with the commercial VASH1 that contains only the 27kDa N-Terminal part of the protein coupled to GST, lacking the main predicted RNA binding C-terminal, domain (see Figure 5 and EV Figure 4). Since last year, we have started a collaboration with Yasufumi Sato (Sendaï, Japan), who sent us the full-length recombinant protein. The experiments with the full-length protein have been done for FGF1, VEGFA and EMCV IRESs. The KDs are 6.5 nM, 8 nM and 9,6 nM, respectively, that reflects a 400 times stronger affinity for FGF1 IRES than with the previous incomplete protein (2.8 mM, with the only protein that was available on the market!!). It is considered that RRM domains of PTB, for example, binds target RNA with a μM affinity. Here we are at the nanomolar level, which is considered the norm for transcription factors binding to DNA. With these data we can affirm that VASH1 full length isoform is an RNA-binding protein with a high affinity for RNA. VASH1 binding to the three IRES indicates that it is not sufficient to activate IRES activity and probably involves other partners in the cellular context. At the moment we have no data about the binding of VASH1 to endogenous FGF1 RNA. Figure 5 has been changed consequently, as well as the text (subsection “Identification of IRES-bound proteins in hypoxic cardiomyocytes reveals vasohibin1 as a new RNA-binding protein”).

4) Figure 6: Please, show a Western blot with and without VASH1 depletion to ensure that VASH1 antibody only detects the expected protein(s). I do not see a perinuclear localization pattern, but rather a cytoplasmic pattern.

We performed capillary Western that quantitatively show the VASH1 knock-down, whose efficiency is 59% (Figure 7). Due to the stability of VASH1 protein (superior to 24 hours as measured in EV Figure 5 of the revised version), it difficult to obtain a more efficient knock down. New (better) images were obtained with a confocal microscope. VASH1 was quantified (new Figure 6). To do that, VASH1 was colored in green in the cytosol and in red in the nucleus, indicating that VASH1 is present in both compartments. In hypoxia picture, a perinuclear staining becomes visible in hypoxia. However, we prefer to conclude only by saying that VASH is both cytosolic and nuclear. Interestingly, VASH1 is localized in nuclear and cytosolic bodies whose number does not increase but whose size significantly increases in cytoplasm upon stress, suggesting that it could be localized in stress granules.

5) Figure 7C: Given that the activity of the various IRESes in the bicistronic assay is weak (2-fold maximum), it is important to ensure that the reduction after VASH1 depletion is significant and is not affected by the variability inherent to the assays. For this reason, rather than showing one representative triplicate experiment and use standard error of the mean to measure significance, it is important to use the full set of biological replicates and show standard deviation. The same could be applied to Figure 4B-D.The difficulty to obtain efficient VASH1 depletion is probably responsible for the weakness of the observed effects. We have reproduced several times the experiments and present three independent experiments for each IRES with all the crude values in EV Table 5. A reproducible significant effect has been obtained for FGF1 and VEGFD IRESs as well as a small effect on EMCV IRES. Maybe with a better knock down we could have obtained stronger data. The new version of Figure set uses the full set of independent experiments and biological replicates (n=9) to calculate the means. EV Table 5 recapitulates all the luciferase ratio values and clearly shows the reproducibility of our results. This has been done also for Figure 4 (EV Table 3). For each IRES and for each conditions n=9.Also regarding Figure 7C, how do the authors explain the increase in VEGFA IRES activity upon VASH1 depletion if VASH1 does not bind to VEGFA IRES? The suggested inhibitory effect of VASH1 is more than discussible.

We agree with the reviewer: our data do not allow to propose an inhibitory effect of VASH1. This point has been withdrawn from the text.

6) How does depletion of VASH1 affect the polysomal distribution of FGF1 and VEFGA mRNAs in hypoxia? Table 2 shows a comparison of polysomes in hypoxia (4 hours) with siVASH1 polysomes in normoxia, while the correct comparison would be siControl polysomes in hypoxia (4 hours) with siVASH1 polysomes in hypoxia (4 hours).

Indeed, we wished to present the data with the siVASH1 in hypoxia, but we were not successful with that experiment because many mRNAs were not detectable. Probably not enough material was used in that experiment. As we had no opportunity to reproduce this fluidigm experiment, we have withdrawn it from EV Table 2: we agree that showing the data only in normoxia does not provide very useful information.

7) The Discussion section is highly speculative. The existence of "waves" of IRES-dependent regulation cannot be supported by the analysis of just 2-4 IRESes representing about 5% of the full set of translationally activated mRNAs under hypoxia. Furthermore, the physiological relevance of the potential regulation of IRES-dependent translation by VASH1 cannot be inferred by the established roles of VASH1, because each factor regulates multiple targets by different mechanisms. In addition, whether nuclear localization of VASH1 has any relevance for the function described here is an open question. The inference on regulons is too far-fetched given that there is only one member of each proposed regulon.

We agree that our discussion was too speculative. Even though these waves may exist, we were technically limited with the variations of the HL-1 sensitivity to hypoxia that can vary in the early times from one experiment to another, due to small variations of their phenotype with passages, with confluence etc. The more they are confluent and beating, the more they are sensitive in early hypoxia but it was really difficult to clearly establish the time of activation. In the kinetics presented in this new version, we see that there is a peak at 6h for FGF1 IRES. Due to these difficulties we encountered we prefer to give up on the distinction between the two waves of 4 hours and 8 hours, and we only mention early hypoxia (between 4 hours and 8 hours) and late hypoxia 24 hours. In addition, we should have analyzed more IRESs to conclude about this. Also, we agree we have not enough IRESs regulated by VASH1 at the moment to say that there is a regulon. This has been modified in the text.

Reviewer #3:Figure 2:While the data show that there are certainly less "heavy" polysomes in the hypoxic cells it would be better if the areas under the curves were calculated so that these data can be presented additionally as a polysome:subpolysome ratio to provide a stronger indication of the degree of initiation changes. This would also allow the extent of variation between experimentation to be observed. In addition, the authors need to carry out Met labelling or use some other measure to show that global protein synthesis rates are decreasing under these conditions. This should be accompanied by western analysis to show changes in for example, 4EBPs, eIF2α.

Indeed, the polysome profile provided in the first version did not show a decrease of the P/M ratio and probably less lysate had been loaded. We have reproduced the experiment, and a correct polysome profile is now shown, with a P/M ratio (calculated from the areas under the curves, ratio of disome and up to monosome) that decreases from 1.55 to 1.40. The translational efficiency is indeed poor in normoxia in these cells (often observed with non cancerous cells), but still decreases in hypoxia. Western blot (capillary electrophoresis) have been performed to detect, 4EBP, phospho4EBP, eIF2a and phospho-eIF2a. Unexpectedly, 4EBP appears as only one band in these cells (probably hypophosphorylated) and under hypoxia there is a small increase of this protein but no change on the level of phosphorylation. In contrast we observe a strong increase in eIF2a phosphorylation. These immunodetections have been added in Figure 2 and EV Figure 2.

Figure 3The data show fold-changes in polysome mRNA localisation and in addition changes in transcription. Since the transcription and translation data have been obtained using arrays and, it is important to show how these transcripts vary across the gradients in hypoxic cells to confirm these findings, for a couple of mRNAs. Is it possible to calculate changes in translational efficiency from these data taking into account the transcriptional variation? This information would be very useful.

The mode of calculation of mRNA in polysome was not clearly explained in the first version. The values in the first version already corresponded to the ratio of mRNA in polysome to total RNA. It is now clearly explained in the text (Mat et meth pages 17 bottom, legends of Figure 2 and EV Table 2). In the new EV Table 2, we provide all the values (total RNA, polysomal RNA and ratio). In addition, we show it for a second fluidigm experiment with two times of hypoxia (4 hours, 24 hours).

Figure 4The data show how the IRES function changes, but it is important to show how these data correlate with changes in expression of the corresponding proteins. For example, is there an increase in VEGF of 1.9 fold after 8 hours of hypoxia? If the half lives of these proteins are too long, pulse IP may need to be used.

We have measured the FGF1 expression in normoxia and hypoxia by capillary electrophoresis. The normalized areas show an 2.5 fold increase of the FGF1 protein in hypoxia.

Figure 6.It is essential to measure the half-life of VASH1. Another explanation for the protein remaining expressed, even though the mRNA is decreasing, is that it is very stable with a long half-life. Therefore, it would not be turned over sufficiently during the course of the experiment to see a real difference.

The half-life of VASH1 was measured and is presented in EV Figure 5, comparatively to a control (P21). VASH1 half-life is superior to 24 hours, which can explain the difficulty to have large differences by knock down. The Western blot (capillary electrophoresis) shows a knock-down of 59% after 48 hours of siRNA treatment). This result is presented in the new Figure 7.

[Editors' note: the author responses to the re-review follow.]

[…] Essential revisions:The three original reviewers of your paper agree that you performed many additional important experiments and that the paper has improved considerably. However, as you can see two reviewers argue that some key issues remain to be addressed before the paper can be recommended for publication in eLife. These concerns were discussed and agreed on by all reviewers. In particular, it is critical to show that protein expression of endogenous VASH1 changes as the paper claims. You need to carry out pulse labeling IP of VASH1 to convincingly show changes in synthesis. You also ought to provide evidence that VASH1 specifically binds to the endogenous FGF-1 IRES. Another important issue that was raised by reviewer #2 concerns the statistical analysis of the data. The reviews of the three referees are attached so that you can read the detailed comments.Reviewer #2:This is a revised manuscript from Hantleys et al., entitled "Vasohibin1, a new IRES trans-acting factor for induction of (lymph)angiogenic factors in early hypoxia". […] They also identify an ITAF, VASH1, that binds to IRESs in the nanomolar range and that FGF1 IRES activity depends on this ITAF during hypoxia.However, there are a few things that reduce enthusiasm such as the Fgf1 western which doesn't look like there is an increase in protein levels under hypoxic conditions.

We performed other capillary Western experiments that show a convincing increase of FGF1 protein expression upon hypoxia. A new picture has been added in Figure 4C.

Also, the IRES activities increase during hypoxia, but only by 1.2 to 1.7-fold. Is this sufficient enough for a biological response? Perhaps this is why the western didn't show a significant change in FGF1 protein levels or perhaps the increase in IRES activity is muted simply due to using immortalized cells or a promoter that overexpresses the dicistronic RNA. It is difficult to say. Even the ITAF, VASH1, which they reported bound to EMCV and FGF1 IRESs but not VEGFA IRES still had similar binding Kd for all of the IRESs. Overall, many of the conclusions are based on small effects or minor discrepancies that weaken the main conclusions, however, it is possible that these small changes in IRES activity are sufficient to alter cellular programs.

Indeed, IRES activities do not increase very strongly upon hypoxia but these increases are significant. Is this sufficient enough for a biological response? The answer is yes if one refers to several previous publications. A paragraph about this issue has been included Discussion section (subsection “Pathophysiological impact of a moderate stimulation of translation”):

“A striking feature of our data is that the stimulation of IRES activities by hypoxia in cardiomyocytes is moderate, only by 1.3 to 1.7 fold. […] Globally, cellular IRESs show lower degree of activation than viral IRESs, as illustrated by Braunstein et al. who report that the HIF1 IRES is stimulated by 1.6 fold during hypoxia, while the VEGFA IRES is stimulated by 2 fold and the EMCV IRESs by 3.5 fold (29).”

The similar Kd of VASH1 for all IRESs indeed means that VASH1 does not bind specifically to a given IRES.

We have discussed the absence of specificity of VASH1 binding in the Discussion section (subsection “VASH1 impact on translational control can be positive or negative”). The data with the fluidigm PCR array show that VASH1 has a wide impact on recruitment into polysomes of more than 60% of the mRNAs detected in the array. This supports a wide binding of VASH1 to RNA. In addition, our hypothesis is that the IRESome is not limited to the interaction of a single protein with RNA, but is multi-partner, explaining that a given ITAF can be either an inhibitor or an activator. This has been shown in several previous reports for at least 10 other ITAFs (we made an update about this in our review article by Godet et al., 2019).

“Among the IRESs analyzed in the present study, the FGF1 IRES is the only one to be strongly regulated by VASH1 in hypoxia. However, VASH1 was also bound to the EMCV IRES in the BIA-MS experiment, and calculation of affinity constants does not reveal significant differences of affinity for FGF1, VEGFA or EMCV IRES. This apparent inconsistency finds an explanation if one considers the effect of VASH1 in normoxia: there is a trend of IRESs to be activated upon VASH1 depletion, significant for VEGFD and EMCV IRESs. Such data suggest that VASH1 binding is probably not specific to a given IRES, but that different VASH1 partners would be recruited in the IRESome and result in positive or negative effects of this ITAF. The hypothesis of a dual role for VASH1 in translational control is confirmed by the effect of VASH1 depletion on translatome: recruitment into polysomes is affected negatively or positively for 60-70% of genes both in normoxia and in hypoxia. Although the RNA binding ability of VASH1 has been clearly shown in the present study, we cannot affirm that VASH1 impact is direct for all of these mRNAs. Nevertheless, a dual role of activator and inhibitor has been reported for more than ten other ITAFs. Our hypothesis thus remains that the key of the regulation of IRES activity by ITAFs is not RNA binding specificity but rather IRESome multi-partner composition (10).”

The authors reference a Figure 3C when there isn't a figure 3C.

Indeed, it has been corrected.

One of the key figures in the paper that supports a major conclusion that IRES activity leads to increased Fgf1 protein levels is not convincing. Figure 4C has no loading control, such as probing for another protein like actin that shouldn't change in amount from well to well (loading equal protein levels is standard but not sufficient). Furthermore, the quantification of the band intensities does not match the image of the bands in that there isn't an almost 2.5-fold increase in the Fgf1 band under hypoxic conditions (16 to 41). Nor does the graph in 4C (right) look like there is a 2-fold change. For example, figure EV 2 shows a 1.24-fold change that is striking so a 2.5-fold increase should very apparent. Overall, this figure was not clearly explained other than it is a western, thus it is not clear what the graph is or how to interpret this panel.

All the “Western” experiments shown in this paper are capillary Western. Each sample is loaded on an individual capillary. The “Jess” device calculates the ratio of a peak of a given protein to the total proteins effectively loaded in the capillary. This is more accurate and quantitative than in the classical western. Additional explanations have been included in figure legends and in the Materials and methods section to explain this. In the following link you can find explanations about Jess: https://www.proteinsimple.com/simple_western_videos.html

The quantification of the bands do not match with the image because the quantification is normalized to the total proteins. However, we agree that the previous image (before normalization) was not convincing for the reader, thus the Jess Simple Western with FGF1 antibody has been run again and the resulting image clearly shows the increase of FGF1 protein.

The hairpin control for the negative control for IRES activity is better than no control but it is not sufficient to rule out other artifacts such as cryptic promoter activity.

The presence of a cryptic promoter has been checked and figures about that included in all our previous papers with these IRESs. The only 5’ UTR to show a promoter activity is the VEGFA, but this promoter has been mapped between the two IRESs (Bornes et al., 2007). See Figure 2 of this paper. The level of each cistron is quantified by RTqPCR and it is clearly visible when there is an internal promoter.

In addition to all the published data, we checked this issue by RT qPCR of LucR and LucF in transduced HL-1 cells. The data have not included in the paper but are shown in Author response image 1 for the FGF1 IRES. These data measure the ΔΔCT compared to EMCV IRES. The ratio of 1 indicated that there is the same amount of the two cistrons, thus no cryptic promoter.

One thing that would be very informative is to know whether the first and second cistron in the dicistronic reporter are in the same reading frame or not. Also, if the hairpin control has the ORFs in the same reading frame. The impression this reviewer gets is that the control is not really identical to the other IRES constructs, but a negative control used for other IRES constructs. Regardless, the hairpin IRES negative control is not explained in the methods. A true negative control would be the same reporter with an insert that didn't have IRES activity.

The bicistronic vector with a hairpin has been fully validated as a negative control in our previous publications (the first one is Creancier et al., 2000). We do not believe that this control can be questioned. We have looked to the ORF of the two cistrons. Indeed, the intergenic region is a multiple of 3 thus the two ORFs are in the same frame. This could generate reinitiation but apparently, we do not observe it. The intergenic region contains two stop codons in the frame. The hairpin is a very stable structure expected to block any reinitiation. Construction of the hairpin vector has been explained in the Materials and methods section and the intergenic sequence has been added in Supplementary file 8A. The palindrome is indicated in color.

taaACTAGACGCGCTCTCCGTGAACTAGCGTAGCTGACCGATATCGGTCAGCTACGCTAGTTCACGGAGAGCGCGACTAGTGGATCCatg

Nuclearized – used in Figure 6 legend and in the results should be changed to "relocalization to the nucleus" as nuclearized means to supply with nuclear weapons or deploy nuclear weapons in (such as outer space).

We are sorry for this mistake. It has been corrected.

The description of "waves" of IRESs is not justified by the data as they did not clearly show that there are multiple IRESs that are activated at distinctly different times. It wasn't clear that there were 2 distinct classes of IRESs that were activated at 4 hours of hypoxia and a different class at 24 hours of hypoxia. Were there examples of IRES that were active at different times? Yes, but EMCV was up at 4 hours and 24 hours if their statistics are correct (Figure 4D). Also, the error bars were pretty large and the differences were not very big.

The term of waves has been removed from the text. With the new statistical analysis, we see that EMCV and c-myc are not induced in early hypoxia. Whereas the other ones are, except for VEGFA which is almost inactive if one looks at the values in Supplementary file 3. The error bars are large because this is standard deviation, and because we have regrouped a lot of independent experiments where the cells did not exhibit an identical sensitivity to hypoxia depending on various parameters. Despite of this, the Mann-Whitney test shows data with p-values between 0.05 and 0.0001.

Reviewer #3:[…] Most importantly, the main conclusion of the paper, that VASH1 is an ITAF, is challenged by the lack of data showing that the protein indeed binds to endogenous IRES-containing mRNAs. The authors show that endogenous VASH1 binds to biotinylated IRESs, and that VASH1 affects the expression of one IRES reporter (that of FGF1), but no data are shown on the effect of VASH1 on endogenous transcripts or products. The effect on IRES reporters other than FGF1 is negligible, even if statistically significant for some. So VASH1 may very well be a functional ITAF for just one of the analyzed transcripts, without a generalized function in IRES-mediated translation or hypoxia.

Due to the VASH1 antibody which did not work in IP, it was not possible to obtain the RIP data despite several attempts. However, we address the issue of VASH1 effect on endogenous mRNAs by analyzing the effect of VASH1 depletion on translatome by fluidigm deltagenes PCR array: there is showing a strong effect on the translatome with 60-70% of mRNAs whose recruitment into polysomes varies positively or negatively (Figure 8, Supplementary file 7). This wide functional impact of VASH1, although not meaning that the interaction is direct (but RIP does not any more prove a direct interaction), strongly suggests that VASH1 interacts with numerous transcripts, directly or indirectly. The direct binding of VASH1 to RNA is clearly shown by the affinity constants measured in Figure 5.

Thus, paper shows that VASH1 is an ITAF only for FGF1 IRES, however we have preliminary data with other IRESs, for example IGF1R whose activity is sensitive to VASH1 depletion in hypoxia. However, we have not yet sufficiently reproduced this experiment (n=3) to publish it (F. David, unpublished). It is however consistent with the fluidigm data with siVASH1 (Supplementary file 7) where the IGF1R mRNA recruitment into polysomes decreases by 3.5 fold in hypoxia.

Essential revisions:1) For the reason stated above, I think the Title and Abstract are over-stated.

We agree. The Title and the end of the Abstract have been changed.

2) The authors conclude that most genes are not induced at the transcriptome level in hypoxic cardiomyocytes, while they are controlled at the polysome level. But the thresholds to consider induction are different in Figure 1 (transcriptomics) and Figure 2C (translatome).

This has been corrected. The same threshold of 1.5, has been used for all the experiments. This new threshold is interesting as it allows to see differences between the different times of transcription. Even with this threshold (the previous threshold was 1), polysome recruitment remains activated for 94% of the genes. The conclusions in the text have been slightly modified following the changes of percentages that mostly concern transcriptome.

3) Regarding Figure 1, an unbiased clustering to detect coordinated behavior of transcripts along hypoxia would be a good addition to the manuscript.

We agree with this comment, but it was technically difficult for us to add this.

4) Figure 4C: The authors state that "IRES induction correlates with an increased expression of FGF1 protein". However, rather than an increase I see very similar bands and quantification peaks in this figure.

Quantification of the bands does not match with the image because it is normalized to the total proteins, that are not represented on the “numerical” blot. However, we agree that the previous image (before normalization) was not convincing, thus the Jess (capillary) Simple Western with FGF1 antibody has been run again and the resulting image clearly shows the increase of FGF1 protein.

5) Figure 4D: The authors keep talking about activation "waves", when there is only one construct corresponding to a cellular mRNA (c-myc) that is activated at 24 hours, and where only a few IRESs (which the authors sustain represent all IRES-containing mRNAs involved in angiogenesis) have been analyzed altogether. This is another over-statement.

We agree. The term of “waves” has been removed from the text. However, our data clearly show that induction of IRES activity for (lymph)angiogenic factor mRNAs occurs in early hypoxia. The new statistical test (Mann-Whitney) confirm that it occurs, except for VEGFA IRES a (which has a very poor IRES activity anyway in these cells).

6) Figure 5: It is now clear that VASH1 binds RNA with high affinity, but the specificity is an issue. in vitro, the protein binds to all tested IRESs, and in the BIA-MS experiments VASH1 was detected 0/5 times bound to VEGFA, 2/5 times bound to FGF1, and 4/5 times bound to EMCV IRESs. The BIA-MS experiments were done in different conditions, and were repeated just once per condition. From these data, the authors conclude that VASH1 shows specificity for FGF1 and EMCV in cellulo. In my view, additional assays are required to conclude that VASH1 shows specificity in cellulo, such as RIP-qPCR using oligos for endogenous mRNAs.

Indeed, our data show (and we conclude with this) that VASH1 does not bind specifically to a given IRES. Due to the VASH1 antibody which did not work in IP, it was not possible to obtain the RIP data despite several attempts. However, the wide effect of VASH1 depletion on recruitment into polysomes of more than 60% of the mRNAs detected in the array does not support a specific binding of VASH1 to RNA. Our hypothesis is that the IRESome is not limited to the interaction of a single protein with RNA, but is multi-partner, explaining that a given ITAF can be either an inhibitor or an activator. Dual ITAF effect has been shown in several previous reports for at least 10 other ITAFs (we made an update about this in our review article by Godet et al., 2019). We discuss this issue in the Discussion section.

7) Figure 6: The authors state that "VASH1 immunodetection confirmed a strong expression of VASH1 at 4 hours of hypoxia, despite the decrease of its mRNA". They have not included these data in the manuscript. It is important that a Western blot showing the induction of the protein at 4hours of hypoxia compared to normoxia is included to back-up the polysomal RNA level analysis of part B.Also in this figure, please use a term other than "nuclearized" to indicate an increase in size of VASH1 foci. This is a confusing term, as it implies the nuclear compartment.

We performed capillary Simple Western and clearly show by that VASH1 is induced at 4 hours. This has been included in Figure 6C. The complete kinetics is included below, with the quantification. We have removed “nuclearized”. Sorry again for this mistake.

**Author response image 2. respfig2:** 

8) Figure 7 shows that depletion of VASH1 affects FGF1 IRES activity under hypoxia. The effect on all other IRESs is negligible even if in some cases statistically significant. Therefore, I find the sentence "These data showed that VASH1 behaves as an activator ITAF in hypoxia, limited to FGF1, VEGFD and EMCV IRESs, while it has an inhibitory role on the activities of these IRESs in normoxia" a strong overstatement.

We have modified the text in subsection “Vasohibin1 is a new ITAF active in early hypoxia”: “In contrast, in hypoxia, VASH1 knock-down resulted in strong decrease of FGF1 IRES activity, by 64%, whereas it did not significantly affect the other IRESs (Figure 7D). These data showed that VASH1 behaves as an activator of FGF IRES in hypoxia, while it tends to inhibit several IRESs in normoxia (Figure 7C).”